# Active site remodeling in tumor-relevant IDH1 mutants drives distinct kinetic features and potential resistance mechanisms

Matthew Mealka[1], Nicole A. Sierra[1], Diego Avellaneda Matteo[1], Elene Albekioni[1], Rachel Khoury[1], Timothy Mai[1], Brittany M. Conley[1], Nalani J. Coleman[1], Kaitlyn A. Sabo[1], Elizabeth A. Komives[2], Andrey A. Bobkov[3], Andrew L. Cooksy[1], Steve Silletti[2], Jamie M. Schiffer[4], Tom Huxford[1] & Christal D. Sohl[1]✉

Mutations in human isocitrate dehydrogenase 1 (IDH1) drive tumor formation in a variety of cancers by replacing its conventional activity with a neomorphic activity that generates an oncometabolite. Little is understood of the mechanistic differences among tumor-driving IDH1 mutants. We previously reported that the R132Q mutant unusually preserves conventional activity while catalyzing robust oncometabolite production, allowing an opportunity to compare these reaction mechanisms within a single active site. Here, we employ static and dynamic structural methods and observe that, compared to R132H, the R132Q active site adopts a conformation primed for catalysis with optimized substrate binding and hydride transfer to drive improved conventional and neomorphic activity over R132H. This active site remodeling reveals a possible mechanism of resistance to selective mutant IDH1 therapeutic inhibitors. This work enhances our understanding of fundamental IDH1 mechanisms while pinpointing regions for improving inhibitor selectivity.

IDH1 is a highly conserved, homodimeric enzyme that reversibly converts isocitrate (ICT) to α-ketoglutarate (αKG) through NADP⁺-dependent oxidative decarboxylation. Tumor-driving IDH1 mutants catalyze a NADPH-dependent conversion of αKG to the oncometabolite D-2-hydroxyglutarate (D2HG), while typically ablating the conventional reaction[1–3]. D2HG competitively inhibits αKG-dependent enzymes like TET2 and JmjC lysine demethylases, causing DNA and histone hypermethylation and cellular de-differentiation[4,5]. Mutations at R132 drive >85% of lower grade and secondary gliomas[6] and ~40% of cartilaginous tumors[7], with R132H typically the most common[8,9]. Mutated IDH1 has been successfully therapeutically targeted, with several FDA-approved selective inhibitors in use and more in clinical trials (reviewed in refs. 10–12).

While early kinetic characterization of IDH focused on bacterial forms, recent efforts have illuminated details of human IDH1. As wild type (WT) IDH1 binds its substrates, a conformational change occurs where the large domain (residues 1–103, 286–414) and small domain (residues 104–136, 186–285) move towards each other owing to a hinge (residues 134–141) within the clasp domain (residues 137–185)[13]. This movement closes the active site cleft with the concomitant opening of a back cleft[13]. In the absence of bound substrates, the α10 helix (residues 271–285) helps stabilize IDH1 in its open, inactive conformation[13]. This critical regulatory element undergoes a conformational change to help properly orient the active site residues upon substrate binding-driven closure[13]. These structural features are generally preserved in IDH1 R132H[1,3,14], but inherent catalytic deficiencies coupled with improved NADPH binding result in this mutant catalyzing D2HG production, albeit inefficiently though at great benefit to the tumor environment.

To better understand how D2HG production occurs, there is tremendous value in studying a variant of IDH1 with more robust

¹Department of Chemistry & Biochemistry, San Diego State University, San Diego, CA, USA. ²Department of Chemistry & Biochemistry, University of California San Diego, La Jolla, CA, USA. ³Sanford Burnham Prebys Medical Discovery Institute, La Jolla, CA, USA. ⁴Vividion Therapeutics, San Diego, CA, USA. ✉e-mail: csohl@sdsu.edu

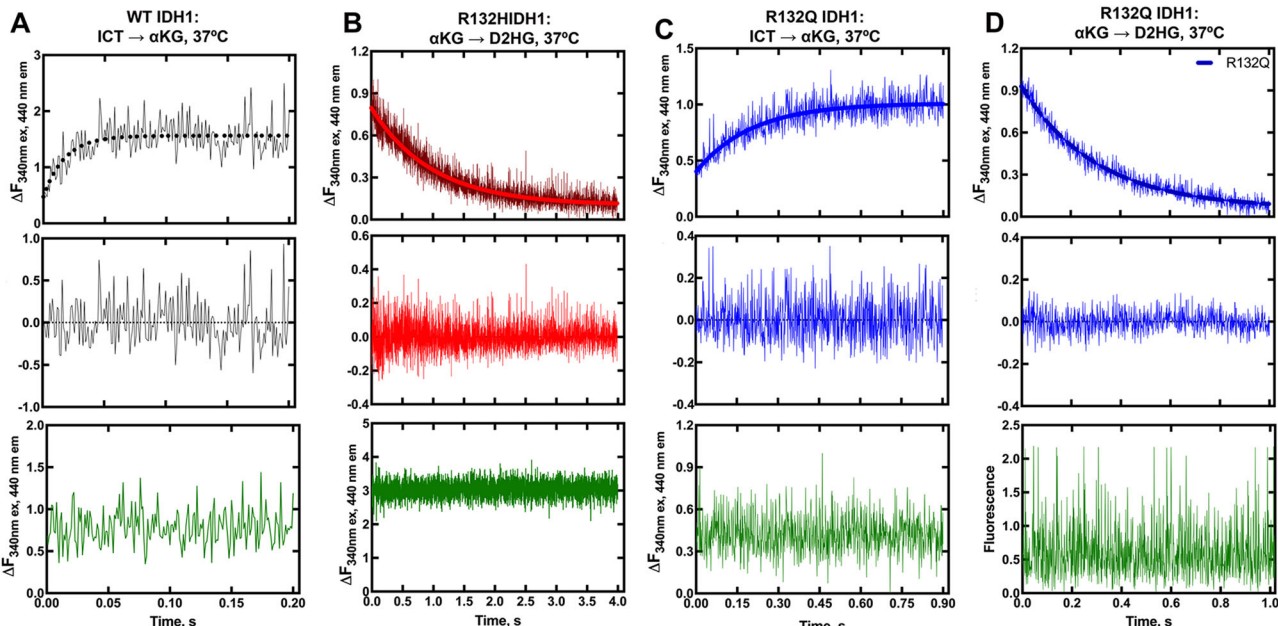

**Fig. 1 | Pre-steady-state single-turnover kinetic features of IDH1 WT, R132H, and R132Q catalysis.** NADPH formation in the conventional reaction and consumption in the neomorphic reaction was monitored over the course of a single turnover (top plot) and compared with a control experiment lacking enzyme (bottom plot, in green). Traces represent an average of four technical replicates. Residuals (middle plot) were obtained to assess the goodness of a single exponential equation fit in the top plots. Kinetic parameters were calculated and reported as +/−SEM resulting from deviation of the mathematical fit. **A** IDH1 WT, conventional reaction. **B** IDH1 R132H, neomorphic reaction. **C** IDH1 R132Q, conventional reaction. **D** IDH1 R132Q, neomorphic reaction.

neomorphic reaction activity. IDH1 R132C/S/L/G/Q mutations have been reported in patients at lower frequencies[15–19] and support distinct tumor D2HG levels[20]. We have demonstrated that these mutants display distinct kinetic profiles for both neomorphic and conventional reactions[21,22], suggesting that their kinetic features may drive some of the variability of patients' D2HG levels[22]. We identified one mutant, R132Q, that maintained weak conventional catalytic activity, drove robust D2HG production[21], and was resistant to mutant IDH1 inhibitors via a mechanism not yet understood[22]. Additionally, IDH1 R132Q drove enchondroma tumor formation in mouse models[23]. By identifying distinct features of R132Q and R132H, we can uncover selectivity handles for improved mutant IDH1 inhibitors, as an H-to-Q mutation requires only a single base change. Investigating the atomic-level mechanisms that drive diverse kinetic activity and inhibition among tumor-relevant IDH1 mutants can also inform chemical features that guide the field of enzyme design[24].

Here, we report the static and dynamic structural features that drive the distinct kinetic properties among tumor-relevant IDH1 mutants, capitalizing on the unusual active site attributes that allow R132Q to maintain conventional and enhance neomorphic activities. We observe by X-ray crystallography that the neomorphic substrate αKG, but not the conventional substrate ICT, binds via multiple conformations to R132Q. Solution-based kinetics and structural experiments demonstrate that ability of R132Q to explore multiple conformations and substrate binding modes depends upon a relatively immobile, solvent-inaccessible enzyme that is better optimized for substrate binding, hydride transfer, and mutant IDH1 inhibitor resistance compared to R132H.

## Results

### R132Q is optimized for substrate binding and catalysis
We previously demonstrated that IDH1 R132Q maintains weak catalytic efficiency for the conventional reaction (ICT to αKG), while also displaying higher catalytic efficiency for the neomorphic reaction (αKG to D2HG) relative to R132H[21,22]. Steady-state kinetics analysis (Supplementary Fig. 1) revealed a 5.9-fold increase in catalytic efficiency for the conventional reaction in R132Q versus R132H, driven primarily by an increase in $k_{cat}$. R132Q catalyzed the neomorphic reaction 9-fold more efficiently than R132H via optimization of both $k_{cat}$ and $K_m$. This suggested that R132Q exhibits a more stable transition state and provides more optimized on/off paths of the reactants and products compared to R132H.

Pre-steady-state kinetics experiments indicated that hydride transfer, or a step preceding it, was rate-limiting for the conventional reaction catalyzed by WT and R132Q, and for the neomorphic reaction catalyzed by R132Q and R132H (Fig. 1). NADPH consumption by R132H showed an initial lag that was eliminated when using higher concentrations of αKG (Supplementary Fig. 2). A lag has been reported previously with IDH1 WT, which was eliminated via pre-incubation of both ICT and metal[25–28]. Interestingly, we did not observe a lag in the neomorphic reaction catalyzed by R132Q, despite using a concentration of αKG that was 10-fold lower than the concentration associated with a lag in R132H. This suggested that αKG is more proficient at driving R132Q from an inactive to an active state compared to R132H, though it was not apparent through these experiments whether this was achieved by a more catalytically primed ground state or a faster conformational change.

We were unable to capture rates of conformational change when monitoring intrinsic protein fluorescence. However, we measured rates of NADPH binding to IDH1 WT, R132H, and R132Q using enzyme that was stripped of cofactor[14] (Supplementary Fig. 3). We found that all three IDH1 proteins displayed single-step binding events, with an NADPH binding on rate ($k_{on}$) that was ~2-fold faster for WT than R132Q, while $k_{on}$ rates for R132H were profoundly slower. We also used isothermal titration calorimetry (ITC) to measure equilibrium binding affinity of NADPH for IDH1 (Supplementary Fig. 4). We found that both mutants exhibited a decrease in $K_d$ compared to WT, suggesting that a slower $k_{off}$ rate drove the improved affinity for NADPH observed for R132H despite the slow $k_{on}$ rate. Taken together, these kinetic data further supported the finding that when compared to R132H, IDH1 R132Q has a lower barrier to adopting the closed, active conformation that is driven by substrate and metal binding.

## R132Q has a less solvent-accessible active site

To illuminate possible mechanisms behind the time-resolved changes exhibited by IDH1 R132Q versus those in WT and R132H, we first used hydrogen/deuterium exchange-mass spectrometry (HDX-MS) analysis. We probed solvent accessibility as indicated by deuterium uptake in the binary IDH1:NADP(H) form, as WT and mutant IDH1 are known to copurify bound to NADP(H)[26,27]. We also measured deuterium uptake upon the addition of substrate (ternary complex, IDH1:NADP(H):ICT/αKG), or upon the addition of substrate and $Ca^{2+}$ (quaternary complex, IDH1:NADP(H):ICT/αKG:$Ca^{2+}$). By far the most substantial change in deuterium uptake for WT, R132H, and R132Q occurred in the quaternary form, indicative of closed, catalytically competent conformations among all enzyme species (Supplementary Fig. 5). This is consistent with previous findings that both substrate (ICT, but also presumably αKG in the neomorphic reaction) and divalent metal binding are required to drive IDH1 into its fully closed, active conformation[25–28]. Deuterium uptake generally showed the following trend: R132H:NADPH:αKG:$Ca^{2+}$ ≫ WT:$NADP^+$:ICT:$Ca^{2+}$ > R132Q:NADPH:ICT:C-$Ca^{2+}$ > R132Q:NADPH:αKG:$Ca^{2+}$ (Fig. 2, Supplementary Figs. 5 and 6), with R132Q overall appearing to have a less structurally dynamic, more closed conformation compared to R132H.

Since our kinetic studies suggested IDH1 R132Q had a lower barrier to achieve the closed conformation compared to R132H, we hypothesized that binary R132Q:NADP(H) would be in a more quaternary-like state. To test this, we compared deuterium uptake among the binary states, predicting that the R132Q:NADP(H) complex would experience less deuterium uptake than R132H:NADP(H). Unsurprisingly, in general the IDH1:NADP(H) form of all three proteins had high deuterium uptake, particularly in the substrate binding pocket, clasp, and dimer interface (Fig. 2, Supplementary Figs. 5–7). As predicted, R132Q:NADP(H) and WT:NADP(H) had the least deuterium uptake overall, while R132H:NADP(H) exhibited, by far, the most uptake. As this suggested that NADP(H)-bound R132Q had a more closed/less mobile conformation compared to R132H, we wondered if the temporal features of our HDX-MS data suggested a faster closing upon substrate binding for R132Q. This would provide one mechanism of the improved catalytic efficiency shown by IDH1 R132Q relative to R132H in the conventional and neomorphic reactions. To address this, we inspected peptides that included residues within 4 Å of bound NADP(H) and ICT/αKG to compare deuterium exchange rates, as uptake plots represent combined exchanged rates for all amides in the peptide, to compare the composition of exchange rates in R132Q versus R132H. We expect that fewer amides would be exchanging at slower exchange rates for R132Q if this enzyme had a primed ground state that reached a closed conformation more easily[29]. Indeed, many active-site peptides had fewer amides with slower/intermediate exchange rates for IDH1 R132Q and WT compared to R132H (Supplementary Fig. 7). Specifically, peptides 210–216 (including catalytic residue K212), 240–253, and 257–267 all showed contributions of amides exchanging at faster rates for R132Q versus R132H. This favors a model where the ground state of R132Q is a more closed conformation that follows a simpler path to a catalytically competent state compared to R132H.

Seeking to pair the dynamic, intermediate-resolution HDX-MS data with static, high-resolution X-ray crystal structures, we report here six crystallographic models representing the structures of IDH1 R132Q: binary IDH1 R132Q bound to NADP(H) (R132Q:NADP(H), PDB 8VHC, PDB 8VH9; R132Q bound to conventional reaction substrates (R132Q:NADP(H):ICT:$Ca^{2+}$, PDB 8VHD); R132Q bound to neomorphic reaction substrates (R132Q:NADP(H):αKG:$Ca^{2+}$, PDB 8VHB), PDB 8VHA, and R132Q bound to a NADP-TCEP adduct (R132Q:NADP-TCEP:$Ca^{2+}$, PDB 8VHE). These structures facilitated comparisons with previously solved IDH1 WT[13] and R132H structures[14,30], including among binary and ICT- and αKG-bound models.

Binary structures of IDH1 R132Q (Fig. 3A) were valuable to help us understand differences among the mutant active sites. While R132Q:NADP(H) showed no major global structural alterations upon alignment with previously solved structures of WT:NADP(H)[13] and R132H:NADP(H)[30], local shifts were observed (Fig. 3B–D). Unsurprisingly, NADP(H)-bound R132Q had the typical open, inactive conformation seen in WT and R132H, with a larger active site cleft and smaller back cleft relative to the quaternary complexes (Supplementary Table 1). These distances in the binary R132Q structure more closely resembled binary WT than R132H, supportive of a more closed, catalytically competent ground state for R132Q. However, R132Q exhibited notable differences compared to WT and R132H binary complexes. In particular, the clasp domain and helices proximal to the substrate and cofactor binding site were shifted, with the α1, α2, α4, α5, and α11 helices adjusted upwards and inwards in R132Q versus WT and R132H, resulting in a similar shift of the NADP(H) molecule itself in dimer-based alignments (Fig. 3B). Importantly, this inward shifting of the α1 helix is a feature of closed, catalytically competent IDH1 conformations. R132Q also contained longer, more intact β strands in the clasp domain, which plays a major role in maintaining the dimer, compared to WT and R132H (Fig. 3C). The fully intact β7 and β8 strands in R132Q were reminiscent of quaternary, fully substrate-bound forms of IDH1 WT and R132Q (vide infra). Consistent with such stable secondary structure, peptides in the β8 strand of R132Q:NADP(H) had lower deuterium uptake than WT:NADP(H) and R132H:NADP(H) (Fig. 2, Supplementary Fig. 6). IDH1 R132Q also maintained an extensive hydrogen bonding network enveloping the NADP(H) molecule; this network was far less robust in R132H (Supplementary Fig. 8). Together, dynamic and static structural data suggested that the IDH1 R132Q active site pocket and surrounding features have greater rigidity and more defined structural features typical of fully-substrate-bound forms of IDH1, suggesting a more catalytically primed state for R132Q:NADP(H) compared to R132H:NADP(H).

## ICT-bound R132Q is in a closed conformation

Here, we also report an ICT-bound quaternary structure of IDH1 R132Q (R132Q:NADP(H):ICT:$Ca^{2+}$, Fig. 4A). Upon alignment with WT:NADP(H):ICT:$Ca^{2+}$[13] (Fig. 4B, C, Supplementary Fig. 9B), there was obvious overlap in both global features and active site details. ICT-bound IDH1 R132Q also aligned well with R132H bound to its preferred substrate, αKG (R132H:NADP(H):αKG:$Ca^{2+}$)[14] (Fig. 4B, C, Supplementary Fig. 9D). Like ICT-bound WT and αKG-bound R132H structures, ICT-bound R132Q adopted a catalytically competent, closed conformation, with ICT maintaining many of the same polar interactions with the protein and divalent ion as observed with WT. This is supportive of our kinetic data showing R132Q's preservation of the conventional activity.

Though alignment of ICT-bound WT and R132Q was strikingly similar (Supplementary Fig. 9B), the 220-fold decrease in catalytic efficiency suggested that maintaining hydrogen bonding features and active site structuring was not sufficient for robust conventional activity in R132Q. Interestingly, ICT was observed only in one monomer of the R132Q quaternary complex, resulting in a shift of the α11 helix and the NADP(H) molecule upward and outward in the ICT-absent R132Q monomer (Fig. 4C), reminiscent of the WT:NADP(H) binary structure (Fig. 3). This lack of active site saturation suggested a lower affinity toward ICT for R132Q versus WT. Though $K_m$ values are not affinity measurements, it is noteworthy that there was a 32-fold increase in $K_m$ when comparing R132Q to WT (Supplementary Fig. 1). To address differences in binding affinity, we again turned to ITC experiments. ICT binding affinity for IDH1 R132H was too poor to be detected, while R132Q exhibited ~170-fold worse affinity for ICT compared to WT (Supplementary Fig. 4). Structural studies provided a possible mechanism for ICT's poor binding to R132H versus R132Q; in contrast to the

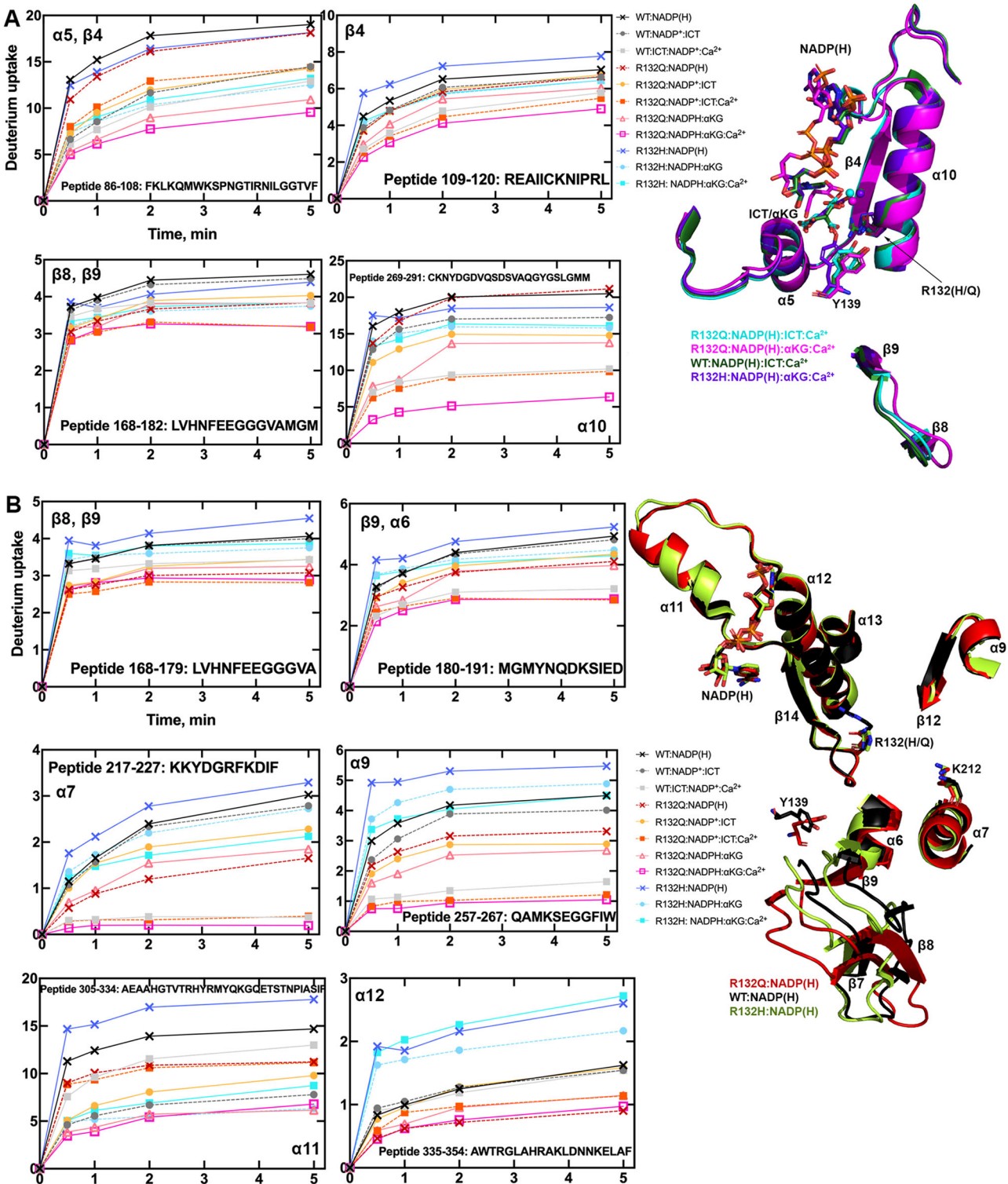

**Fig. 2 | IDH1 R132Q has lower deuterium uptake than R132H in both binary and quaternary complexes. A** Plots of deuterium uptake encompassing residues 86–120, 168–182, and 269–291 (left) are shown with the structural features of these residues shown in cartoon (right) for IDH1 R132Q, WT[13], and R132H[14]. **B** Plots of deuterium uptake for residues 168–191, 217–227, 257–267, and 305–354 (left) are shown, with the structural features of IDH1 R132Q, WT[13], and R132H[30] encompassing these regions indicated in cartoon (right). Each point represents the mean of three technical replicates.

closed, catalytically competent conformation of ICT-bound R132Q, a previously solved ternary R132H:NADP(H):ICT[30] structure revealed quasi-open monomers that had α4 and α11 helices shifted upwards and outwards from the dimer interface and an unraveled α10 helix (Fig. 4B, Supplementary Fig. 9C), regions we and others have shown to be highly flexible[13,30–32]. Notably, ICT was found in a posited pre-

binding site that was shifted to the left of its catalytically-competent position (Fig. 4C)[30]. This resulted in limited polar interactions by ICT to R132H[30] in contrast to ICT's extensive polar contacts to R132Q, including hydrogen bonding to catalytic residue Y139 in R132Q that indicated a catalytically-ready binding conformation (Supplementary Fig. 8). As further evidence that ICT-bound R132H was

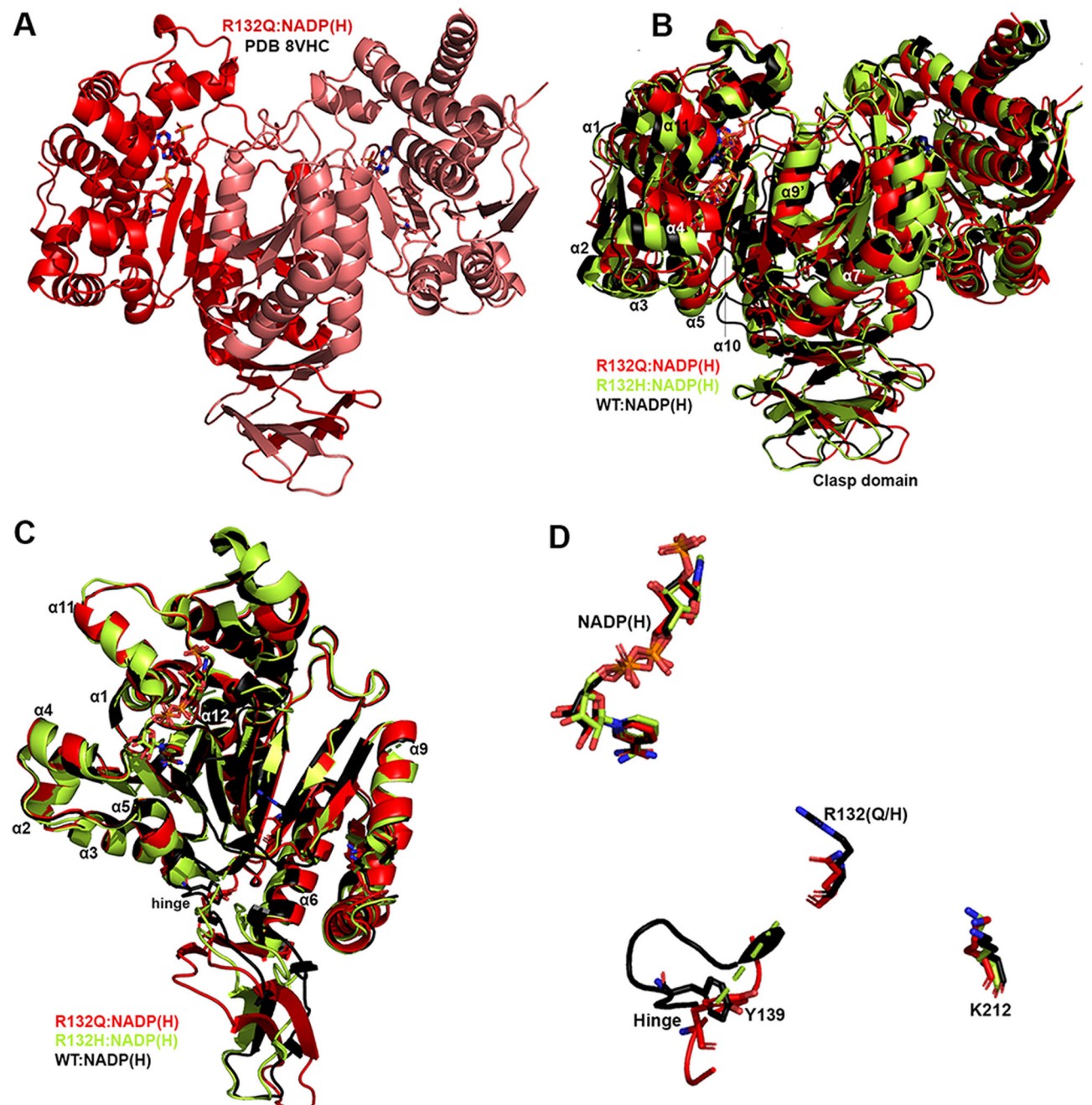

**Fig. 3 | Crystal structure of NADP(H)-bound IDH1 R132Q shows a typical open conformation. A** The binary R132Q:NADP(H) complex is shown with each monomer highlighted using a slight color change. **B** Dimer-based alignments of R132Q:NADP(H) (red), WT:NADP(H)[13] (black), and R132H:NADP(H) (light green)[30]. **C** Monomer-based alignments of the structures in (**B**). **D** The view show in (**C**) was simplified to highlight catalytic residues Y139 and K212 (though the latter residue drives catalysis in the monomer not shown as this is a monomer-based alignment), residue R132(H/Q), and the cofactor.

ill-prepared for catalysis, its catalytic residues were swung away from the active site (Fig. 4C), akin to the positioning found in binary, catalytically incompetent IDH1 structures. Though this R132H structure did not include a divalent metal that may be required for full closure[30], it is nonetheless unsurprising that R132H, in contrast to R132Q, is essentially unable to convert ICT to αKG.

**αKG-bound R132Q has a shifted binding pocket**

Since IDH1 R132Q distinctly maintains both conventional and neomorphic catalytic abilities, we asked how the binding conformations for ICT, the conventional reaction substrate, and αKG, the neomorphic reaction substrate, compared. Here, we report two αKG-containing

R132Q quaternary structures (R132Q:NADP(H):αKG:Ca$^{2+}$). These co-crystallization experiments led to a variety of complexes, with monomer asymmetry observed (Fig. 5). One structure contained a dimer that had αKG and a covalent NADP-αKG adduct bound in its monomers (Fig. 5A). Cleft measurements in both monomers indicated a slightly more open conformation when compared to the closed quaternary R132Q (ICT-bound), WT (ICT-bound) and R132H (αKG-bound) structures, with the α11 helix shifted out away slightly from the substrate binding pocket (Fig. 5, Supplementary Fig. 10). As a result, the NADP(H) itself shifted outwards compared to the ICT-bound R132Q structure, resulting in a semi-closed conformation (Fig. 5D, Supplementary Table 1).

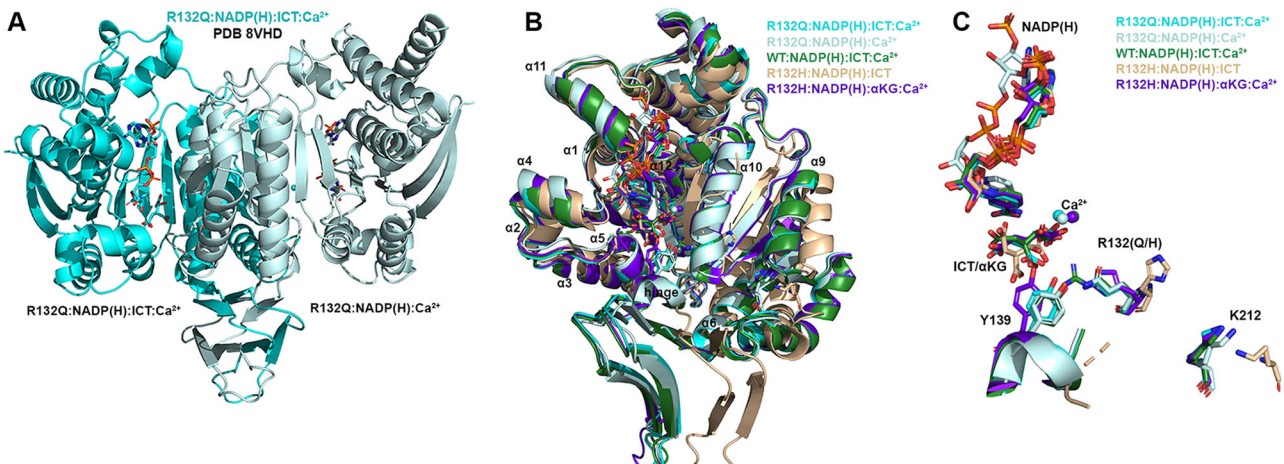

**Fig. 4 | IDH1 R132Q bound to ICT, NADP(H) and Ca²⁺ is in a closed, catalytically competent conformation. A** The R132Q:NADP(H):ICT:Ca²⁺ complex is shown with each monomer highlighted using a slight color change. **B** Monomer-based alignments of R132Q:NADP(H):ICT:Ca²⁺/R132Q:NADP(H):Ca²⁺ monomers (dark and light cyan) with WT:NADP(H):ICT:Ca²⁺[13] (dark green); R132H:NADP(H):ICT[30] (wheat); and R132H:NADP(H):αKG:Ca²⁺[14] (dark purple). **C** For clarity, only the catalytic residues, residue R132X, cofactor, substrates, Ca²⁺ and hinge are shown in the same orientation for the structures shown in (**B**).

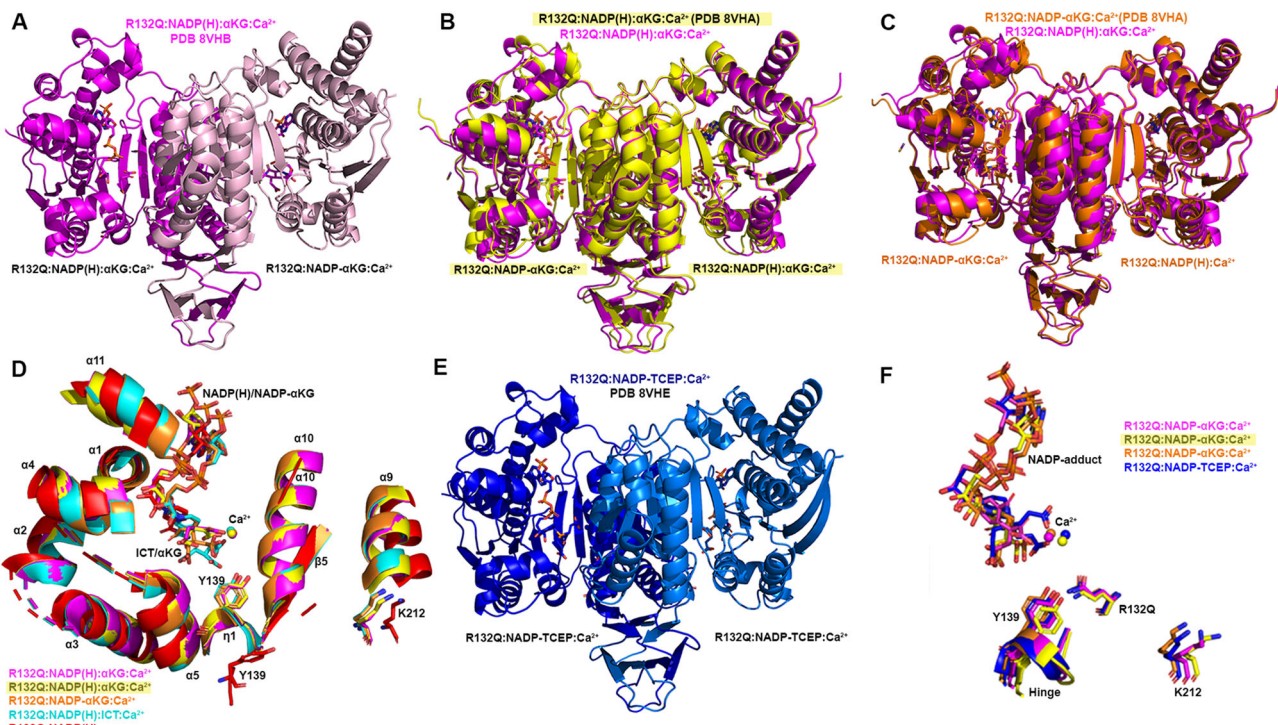

**Fig. 5 | Crystal structure of IDH1 R132Q bound to αKG and NADP-adducts are in closed and semi-closed conformations.** In (**A–C**) and (**E**), a description of the ligands present is listed below each monomer. **A** R132Q:NADP(H):αKG:Ca²⁺/R132Q:NADP-αKG:Ca²⁺ dimer. Each R132Q monomer is highlighted using a slight change in color. **B** R132Q:NADP-αKG:Ca²⁺/ R132Q:NADP(H):αKG:Ca²⁺ dimer 1 (yellow) aligned with the dimer shown in (**A**) (magenta). **C** R132Q:NADP-αKG:Ca²⁺/R132Q:NADP(H):Ca²⁺ dimer 2 (orange) aligned with the dimer shown in (**A**) (magenta). **D** Monomer-based alignment of ICT- and αKG-containing R132Q monomers. **E** R132Q:NADP-TCEP:Ca²⁺/R132Q:NADP-TCEP:Ca²⁺ dimer. **F** Monomer-based alignment of adduct-containing R132Q monomers.

A second αKG-bound structure had distinct features among two dimers in the crystallographic asymmetric unit. One catalytic dimer contained one NADP-αKG adduct and one αKG molecule (Fig. 5B), and again appeared as an intermediate between the R132Q:NADP(H) and the R132Q:NADP(H):ICT:Ca²⁺ structures (Supplementary Table 1, Supplementary Fig. 10). A second dimer contained an NADP-αKG adduct in one monomer, and no αKG-containing molecule in the other monomer (Fig. 5C). This dimer was in a more closed, catalytically competent conformation, reminiscent of the fully closed WT quaternary structure

(Supplementary Table 1, Supplementary Fig. 10). The Ca²⁺ ion clearly led to extensive restructuring, as the R132Q:NADP(H):Ca²⁺ monomer aligned relatively poorly with the R132Q:NADP(H) complex despite the only difference being the metal ion (Supplementary Fig. 10G). Thus, closing of R132Q to the αKG-bound form may be driven just as much by metal binding as by substrate binding. This finding was recapitulated by the overall decrease seen in deuterium uptake upon treatment of substrate-bound R132Q with Ca²⁺ (Supplementary Fig. 5). Overall, we were able to capture snapshots of stable conformations of αKG

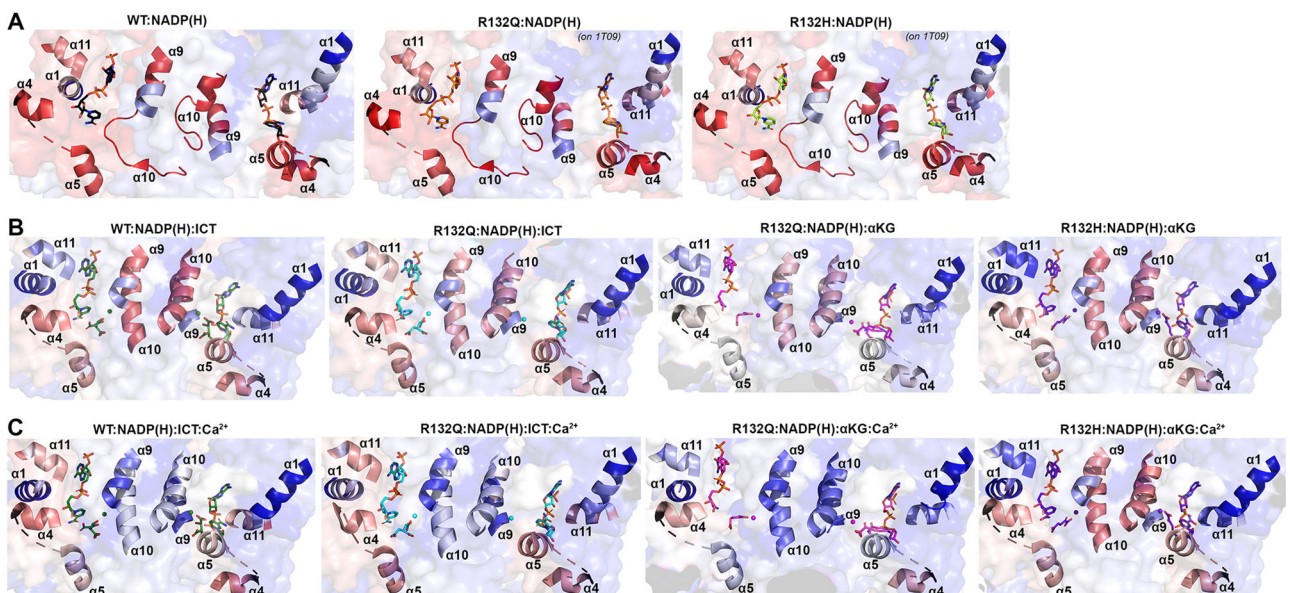

**Fig. 6 | Deuterium uptake by IDH1 WT, R132Q, and R132H in helices bounding the substrate binding pocket.** Deuterium uptake is shown as a gradient from red (high uptake) to blue (low uptake). **A** Deuterium uptake by IDH1 WT, R132Q, and R132H upon no ligand treatment. These HDX-MS data were overlaid on NADP(H)-only bound forms of WT[13] in all three cases, as the αKG helix was disordered in the NADP(H)-only bound forms of IDH1 R132Q and R132H[30]. **B** Deuterium uptake by WT and R132Q upon treatment with NADP+ and ICT, and by IDH1 R132Q and R132H upon treatment with NADPH and αKG. These HDX-MS data were overlaid on WT:NADP(H):ICT:Ca2+[13], R132Q:NADP(H):ICT:Ca2+ and R132Q:NADP(H):αKG:Ca2+, or R132H:NADP(H): αKG:Ca2+[14]. **C** Deuterium uptake by IDH1 WT and R132Q upon treatment with NADP+, ICT, and Ca2+, and by IDH1 R132Q and R132H upon treatment with NADPH, αKG, and Ca2+. These HDX-MS data were overlaid on the structures described in (**B**).

binding ranging from semi-closed (αKG-bound) to essentially fully closed (NADP-αKG adduct-bound).

Closed conformations are seen for WT[13] and R132H[14] when bound with their preferred substrates (ICT and αKG, respectively). As αKG-bound R132Q was often not as fully closed as the ICT-bound form, we wondered how αKG-bound R132Q compared to these WT and R132H closed conformations. In alignments of R132Q:NADP(H):αKG:Ca2+ with quaternary WT and R132H structures, the catalytic residue Y139 in R132Q was shifted away from the αKG molecule (Supplementary Fig. 10), with this molecule making fewer hydrogen bond contacts within the R132Q active site compared to R132H (Supplementary Fig. 8). In R132Q, the αKG binding site was shifted upwards towards NADP(H) and away from the substrate binding sites seen in the ICT-bound WT and αKG-bound R132H structures. This shift might be facilitated by one surprising feature of all non-αKG-containing R132Q monomers -- the nicotinamide ring could not be reliably modeled due to missing electron density. This suggests that when αKG was absent (such as in the R132Q:NADPH:Ca2+ monomer in Fig. 5C) or, more unexpectedly, even when αKG was bound (R132Q:NADPH:αKG:Ca2+ monomers), this portion of NADP(H) was more dynamic. Since the αKG-containing R132Q structures did not appear in a catalytically-ready form, it is possible that the enzymatic mechanism may rely on different amino acids used in the conventional reaction, or, since αKG serves as a substrate and product for R132Q, we may have a view into a product-bound conformation.

### ICT-bound and αKG-bound R132Q are structurally distinct
We found that the α10 regulatory segment underwent the expected notable restructuring upon substrate binding, with this segment forming a helix in both the ICT- and αKG-bound quaternary forms of R132Q (Fig. 5D), just like in ICT-bound WT and αKG-bound R132H. However, our HDX-MS experiments captured more subtle differences in R132Q that depended on which substrate was bound. The α10 regulatory segment and nearby α9 helix were more protected from deuterium exchange in both αKG and αKG + Ca2+ conditions in R132Q than

in the ICT and ICT + Ca2+ conditions (Figs. 2 and 6). Beyond its proximity to the regulatory segment, the α9 helix has an additional role in active site remodeling in that it helps form a "seatbelt" enveloping the NADP(H) cofactor (reviewed in ref. 33). This seatbelt was observed in the WT:NADP(H):ICT:Ca2+ quaternary structure[13], with residue R314 in the α11 helix shifted inward to form polar contacts with D253' and Q256' in α9 of the adjacent monomer and with a water molecule (Fig. 7). The absence of the seatbelt was not limited to binary R132Q, R132H, and WT structures; no seatbelt was observed in the ternary ICT-bound or, more surprisingly, in the closed, quaternary αKG-bound R132H structures[14,30]. As no αKG-bound WT structure is available at this time, we compared a structure of a non-R132 mutant, G97D, which generates D2HG but exhibits a high degree of structural similarities with IDH1 WT[14]. The αKG-bound form of G97D also did not show a seatbelt conformation, suggesting this is a distinct feature of ICT-bound, fully closed structures.

IDH1 R132Q behaved like WT (Fig. 7A) when binding the conventional reaction substrate (ICT), with a seatbelt forming over the cofactor since residue R314 was in position to contact Q256', D253', and, distinct in this protein, E247' in β11 of the adjacent monomer, as well as a water molecule (Fig. 7B). However, R132Q behaved more like R132H (Fig. 7C) when binding the neomorphic substrate, with αKG-bound monomers showing residue R314 swung away from the α9' helix, precluding the necessary polar contacts (Fig. 7B). Interestingly, the closed R132Q:NADP-αKG:Ca2+/R132Q:NADP(H):Ca2+ dimer (Fig. 5C) had an intact seatbelt over the NADP-αKG adduct (Fig. 7B), suggesting that a fully closed conformation of αKG-bound R132Q is possible if the nicotinamide ring of NADP(H) is stabilized in some way, such as via adduct formation. Interestingly, HDX-MS dynamics showed seatbelt formation was associated with an increase in deuterium uptake, with the α11 helix, which contains the seatbelt-forming R314 residue, being more protected in the αKG-bound R132Q and R132H (seatbelt-lacking) complexes relative to the ICT-bound WT and R132Q (seatbelt-forming) complexes (Fig. 6). We note that all of these mutant structures (both R132H and R132Q) describe mutant:mutant homodimers; as these

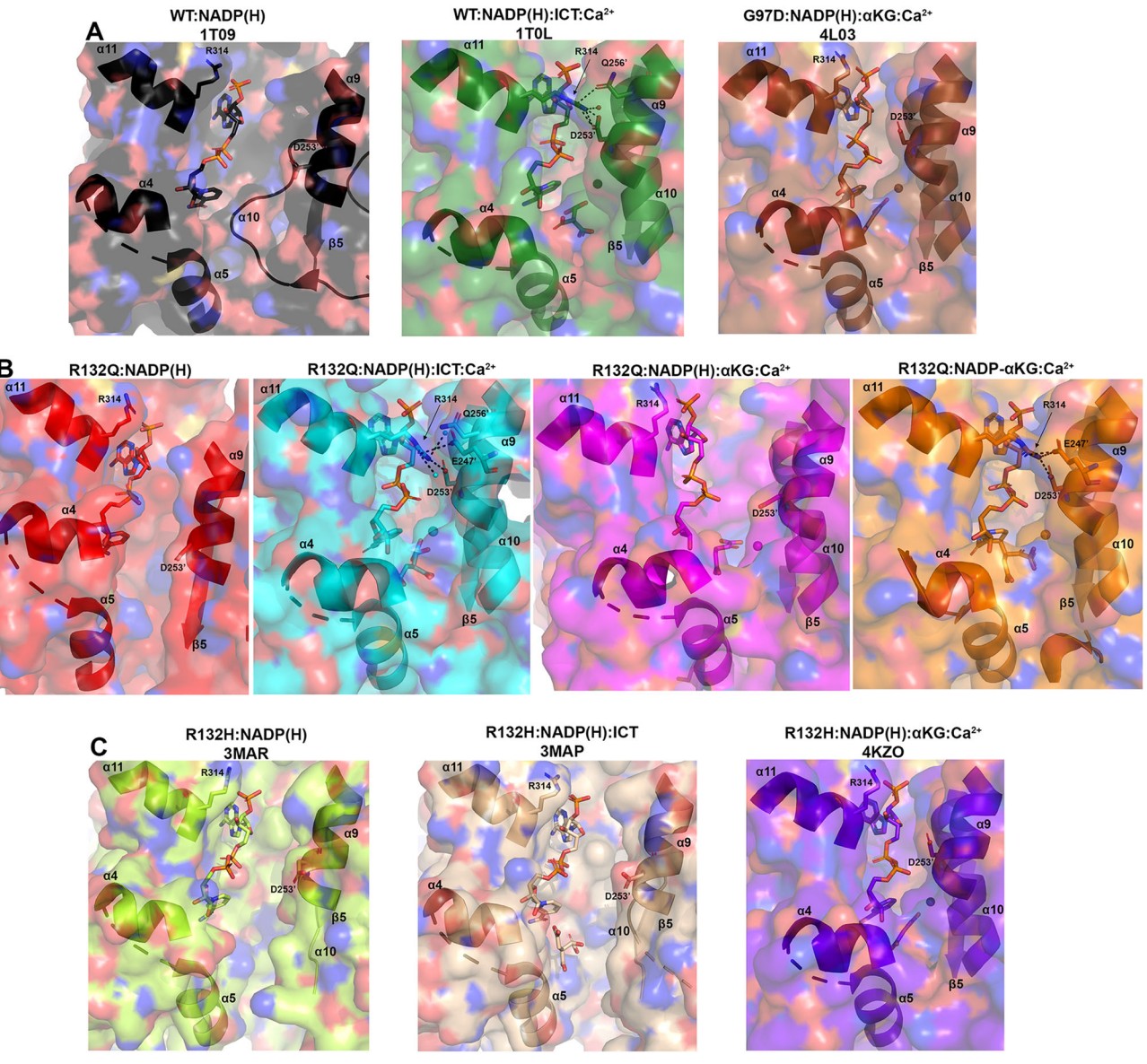

**Fig. 7 | Hydrogen bond network facilitates a "seatbelt" that overlays NADP(H) in only some quaternary structures of IDH1. A** Unlike the binary structure of IDH1 WT[13] and quaternary structure of G97D:NADP(H):αKG:Ca²⁺[14], the quaternary IDH1 WT complex[13] forms a seatbelt over the NADP(H). **B** Binary R132Q:NADP(H) and quaternary R132Q:NADP(H):αKG:Ca²⁺ structures do not form a seatbelt, while R132Q:NADP(H):ICT:Ca²⁺ and the most closed conformation of R132Q:NADP-αKG:Ca²⁺ form a seatbelt. **C** No seatbelt is formed in the binary R132H:NADP(H), ternary R132H:NADP(H):ICT, or quaternary R132H:NADP(H):αKG:Ca²⁺ structures of IDH1 R132H[14,30].

mutations are found heterozygously in patients, a possibility exists for WT:mutant heterodimers, which could result in still different structural features and conformations. Overall, multiple conformations were possible with αKG-containing R132Q structures, including those associated with fully closed forms.

**R132Q accommodates multiple NADP-containing adducts**
In addition to the NADP-αKG adduct, we encountered an NADP-tris(2-carboxyethyl)phosphine (NADP-TCEP) adduct when attempting to crystallize ICT-bound R132Q (Fig. 5E, Supplementary Fig. 11). There may be catalytic relevance to these adducts since the TCEP and αKG carboxylates helped coordinate Ca²⁺ and maintained many hydrogen bonds in their respective active sites, though the metal ion was slightly shifted to accommodate these adducts (Supplementary Figs. 10, 12). All TCEP and αKG adducts appeared as hybrids between the semi-closed, αKG-bound and fully closed, ICT-bound R132Q complexes (Supplementary Table 1). In general, one NADP-αKG adduct-containing

monomer (Fig. 5C) aligned well to the fully closed ICT-bound R132Q structure in all regions except the clasp domain, where the adducted monomer was shifted towards the dimer interface and the β9 strand was more intact (Fig. 5, Supplementary Fig. 10). As further evidence of its fully closed conformation, this NADP-αKG adduct-containing monomer also had an intact seatbelt (Fig. 7B).

To better understand how these adducts were forming, we performed density functional theory (DFT) calculations for model NADP-TCEP and NADP-αKG adducts (Supplementary Tables 2 and 3), which suggested that adduct formation would not occur if not for the constraining environment of the crystal structure. We considered an alternative possibility that the R132Q active site favored adduct formation and binding. If the NADP-TCEP adduct could form in the R132Q active site, it would act as a competitive inhibitor. Thus, we treated R132Q with varying concentrations of three reducing agents (TCEP, dithiothreitol (DTT), and β-mercaptoethanol (BME)) to determine the effects of conventional reaction catalysis (Supplementary Fig. 13,

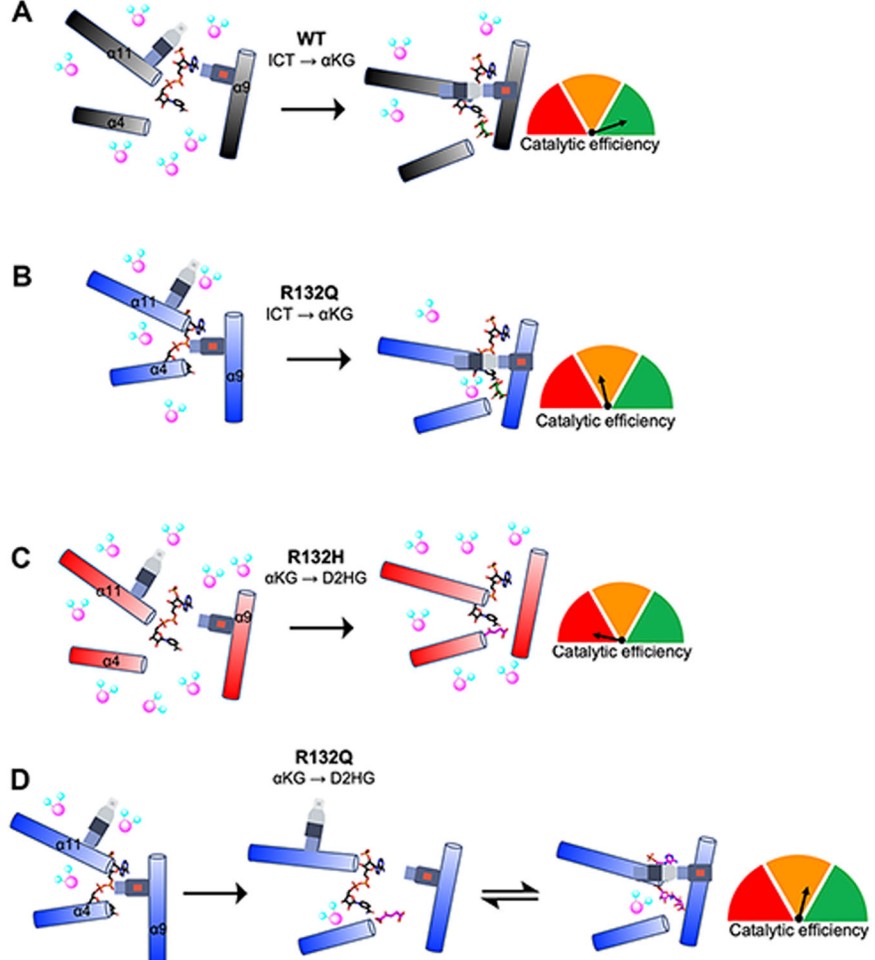

**Fig. 8 | Conformations and solvent accessibility of IDH1 WT, R132Q, and R132H upon substrate binding.** Helices displaying profound differences in alignment of the three forms of IDH1 are highlighted. The seatbelt feature is indicated on the α11 and α9 helices. **A** Binary WT:NADP(H)[13] collapses to a closed conformation upon ICT binding, though moderate levels of deuterium exchange are still permitted. **B** Binary R132Q:NADP(H) collapses to a closed conformation upon ICT binding, showing improved catalytic efficiency for the conventional reaction and lower deuterium uptake compared to R132H. However, catalytic activity is much lower

compared to WT. **C** Binary R132H:NADP(H)[30] collapses to a fully closed conformation only upon αKG binding[14], but a seatbelt is not formed and deuterium uptake remains high. **D** Binary R132Q:NADP(H) forms semi-closed and closed conformations upon binding αKG and NADP-αKG, respectively, with a seatbelt successfully formed in the closed state in some of our crystallographic snapshots. The αKG binding site was shifted away from the α9 helix, though catalytic activity was much higher than that seen in R132H.

---

Supplementary Table 4). Dose-dependent inhibition of R132Q catalysis was profound with TCEP, while DTT and BME had minimal effects. More modest, though notable effects on catalysis were also observed when challenging WT with the highest concentration of TCEP tested (10 mM, Supplementary Fig. 14). Together, these results support the hypothesis that adduct formation occurred outside of the non-physiologically-relevant crystal packing environment, with the adducts mimicking αKG binding, ICT binding, or transition between the two.

As these adduct-containing structures showed hybrid binding features of αKG and ICT, we wondered if transition state features could be extrapolated. Here, the nicotinamide ring of the adduct lent an interesting clue. Calculations suggested that the nicotinamide ring is likely planar in the oxidized form[34,35]. During NADP+ activation for hydride transfer, the enzyme is predicted to distort the nicotinamide ring to form a puckered transition state as a partial positive charge on C4N develops[34–36] (Supplementary Fig. 15). NAD(P)-adducts with reducing agents have been reported previously, including with TCEP[37] and DTT[38], and were found to have a more puckered nicotinamide ring, reminiscent of a transition state.

Here, unlike the planar ring observed in our non-adducted forms of NADP(H) (R132Q:NADP(H):ICT:Ca$^{2+}$), both the αKG- and TCEP-containing NADP-adducts showed a more puckered nicotinamide ring (Supplementary Fig. 12, Supplementary Table 3), suggestive of a transition-state-like conformation.

In summary, we highlight discrete catalytic and structural features among two tumor-relevant IDH1 mutants, with the R132Q mutant serving as an invaluable tool to probe the journey through substrate turnover of two reactions that typically cannot be performed by the same enzyme. Together, our kinetics experiments and static and dynamic structural data suggested that substrate binding and conformational changes associated with the conventional and the neo-morphic reactions have distinct paths through turnover that can be described in terms of differences in substrate affinity, substrate binding site location, solvent accessibility, and propensity for conformational activation and active site remodeling (summarized in Fig. 8). IDH1 R132Q's accommodation of catalytically-relevant adducts, perhaps due to its active site appearing better optimized for catalysis compared to R132H, illuminate snapshots of substrate and substrate analogs in varying degrees of catalytic readiness.

## Discussion

Kinetic, HDX-MS, and crystallography experiments revealed fundamental differences in the catalytic mechanisms of WT and tumor-relevant IDH1 mutants (Fig. 8). We were surprised to identify adducts binding to and, in the case of the TCEP adduct, inhibiting R132Q. While we do not expect this adduct to form under physiologically conditions, our experiments measuring reducing agent inhibition were supportive of the possibility of these adducts representing catalytically meaningful conformations. Determining that the NADP-TCEP adduct was competitive with ICT was unsurprising as the TCEP portion of the adduct mimicked features of ICT binding to R132Q (Supplementary Fig. 12). While this experiment itself does not differentiate whether the adduct forms outside the protein and then binds, or whether the active site pocket drives adduct formation, our DTT experiments may support a model where the enzyme drives adduct formation. NADP-DTT adducts have been previously reported with yeast xylose reductase[38]. While DTT has some similar structural features compared to ICT, it does not recapitulate the carboxylates that the TCEP and αKG adducts contain. These adducts preserved many polar contacts that non-adducted NADP(H) and ICT form with $Ca^{2+}$ and active site residues. Further supportive of this adduct being distinct to R132Q, and thus perhaps formed with the help of this enzyme, TCEP was much less effective at inhibiting WT (Supplementary Fig. 14). This work also highlights a liability for using TCEP as a reducing agent in kinetic and structural studies on dehydrogenases, as adduct formation with $NAD(P)^+$ may complicate structure/function analysis.

We reported previously that IDH1 R132Q has distinct catalytic profiles for the conventional and neomorphic reactions compared to more common tumor-driving IDH1 mutants (R132H, R132C)[21,22]. Though both mutations retain a polar amino acid, a mutation to a glutamine versus a histidine would be expected to be less disruptive due to a more similar size and shape relative to arginine, though it is unsurprising that WT is far more efficient at catalyzing the conventional reaction than the mutants since R132 coordinates the C3 carboxylate of ICT[3,13]. As neither mutant can directly participate in this coordination with ICT, we asked why the conventional reaction was more efficient in R132Q than R132H. We found that R132Q employed an active site water that mitigated the loss of hydrogen bonding to ICT resulting from the R to Q mutation by imperfectly mimicking the polar interactions with the substrate normally afforded by R132 (Supplementary Fig. 16). Despite the shifting of the αKG binding site, we noticed a similar compensatory mechanism in our αKG-bound R132Q structure, with a water molecule again recapitulating these polar interactions. Here, however, the water molecule did not appear to hydrogen bond with the substrate. Instead, a second water molecule was found at the same location as the $Ca^{2+}$ ion in the quaternary ICT-bound WT IDH1 structure (Supplementary Fig. 16A), which presumably helped stabilize the αKG substrate in R132Q. We previously reported the importance of water molecules in facilitating mutant IDH1 inhibition[31], and this current work highlights the importance of water in substrate binding by providing a possible mechanism by which R132Q is more catalytically efficient compared to R132H.

In addition to affecting catalysis, the α10 regulatory segment may serve as a selectivity filter for mutant IDH1 inhibitor binding[39]. We have shown previously that selective mutant IDH1 inhibitors bind poorly to R132Q, with IC$_{50}$ profiles consistent with WT rather than R132H[22]. We predicted that a more stable α10 regulatory segment in R132Q:NADP(H) drove this resistance. However, here we found that while this unfolded loop indeed had stronger electron density compared to R132H:NADP(H)[30], it still appeared less stable than the partially folded features of WT:NADP(H)[13]. We now believe that the more activated, quaternary-like state of the binary R132Q:NADP(H) complex helped drive inhibitor resistance. In this complex, regions including the α11 and α4 helices were shifted inwards, with R132Q experiencing less deuterium uptake (Fig. 8). Using compound 24 as a prototypical

selective mutant IDH1 inhibitor[40], the small increase in the stability of the α10 regulatory segment in R132Q did not appear to have much effect on inhibitor binding (Fig. 9A). Instead, our alignments showed residues 111-121 in the inhibitor binding pocket, which form a loop between the β4 and β5 strands, likely had a larger role in the loss of affinity towards inhibitors for R132Q. While this region accommodated the inhibitor in the R132H:NADP(H) complex (Fig. 9D), these residues interfered with inhibitor binding to R132Q:NADP(H) (Fig. 9C). Interestingly, unlike in R132Q, these residues didn't appear to preclude inhibitor binding in WT:NADP(H) (Fig. 9E). Thus, while the α10 regulatory segment likely precludes inhibitor binding in WT, residues 111-121 perform this function in R132Q (Fig. 9A). Thus, the essentially kinetically identical inhibitory characteristics of WT and R132Q[22] develop through two very different mechanisms. Importantly, as this loop would not have been readily apparent as a selectivity gate when only examining the WT structure, it is only through our R132Q:NADP(H) structure that we were able to identify a possible resistance strategy and selectivity handle.

While much effort has been devoted to understanding the distinct catalytic and structural features of IDH1 WT versus R132H, our discovery of the unusual kinetic properties of the R132Q mutant allowed a valuable opportunity to establish the static and dynamic structural adjustments required to maintain conventional and neomorphic activities within the same active site. Compared to R132H, our findings show that the R132Q binding pocket and surrounding areas are better primed for substrate binding and hydride transfer steps. Rather than simply acting as a hybrid of WT and R132H, R132Q employed distinct strategies to yield improved catalytic parameters for both ICT and αKG turnover as compared to R132H. These structural and dynamic discoveries not only highlight mechanistic properties of important tumor drivers, but also identify regions that may serve as selectivity handles when designing mutant IDH1 inhibitors requiring increasing selectivity or optimization against resistance mutants.

## Methods

### Reagent and tools

Dithiothreitol (DTT), isopropyl 1-thio-β-D-galactopyranoside (IPTG), Triton X-100, α-ketoglutaric acid sodium salt (αKG), DL-isocitric acid trisodium salt hydrate, and magnesium chloride ($MgCl_2$) were obtained from Fisher Scientific (Hampton, NH). BME was obtained from MP Biomedicals (Santa Ana, CA). β-Nicotinamide adenine dinucleotide phosphate reduced trisodium salt (NADPH), β-nicotinamide adenine dinucleotide phosphate disodium salt ($NADP^+$) and tris(2-carboxyethyl)phosphine) (TCEP) was purchased from Millipore Sigma (Burlington, MA). Nickel-nitrilotriacetic acid (Ni-NTA) resin was obtained from Qiagen (Valencia, CA). Stain free gels (4–12%) were obtained from Bio-Rad Laboratories (Hercules, CA). Protease inhibitor tablets were obtained from Roche Applied Science (Penzberg, Germany). Phenylmethylsulfonyl fluoride (PMSF) salt was purchased from Thermo Scientific (Waltham, MA).

### Purification of IDH1 WT and mutant

The *E. coli* BL21 Gold DE3 strain was used for all protein expression. Human IDH1 WT, R132H, and R132Q homodimers were expressed from a pET-28b(+) plasmid in E. coli BL21 Gold DE3 cells. The R132Q construct was made in the WT IDH1 background using site-directed mutagenesis with the following primers: forward primer, 5-GTTAAACCGATCATTATTGGTCAGCATGCCTATGGTGATCAGTATC; reverse primer, 5-GATACTGATCACCATAGGCATGCTGACCAATAAT GATCGGTTTAAC. As described previously[21], incubation in 0.5-1 L of terrific broth with 30 μg/mL of kanamycin (37 °C, 200 rpm) occurred until reaching an A$_{600}$ of 0.9–1.2. Protein expression was induced with 1 mM IPTG after briefly cooling to 25 °C. Following 18 h of incubation (19 °C, 130 rpm), cell pellets were harvested and resuspended in lysis buffer (20 mM Tris pH 7.5 at 4 °C, 500 mM NaCl, 0.1% NaCl, 0.1% Triton

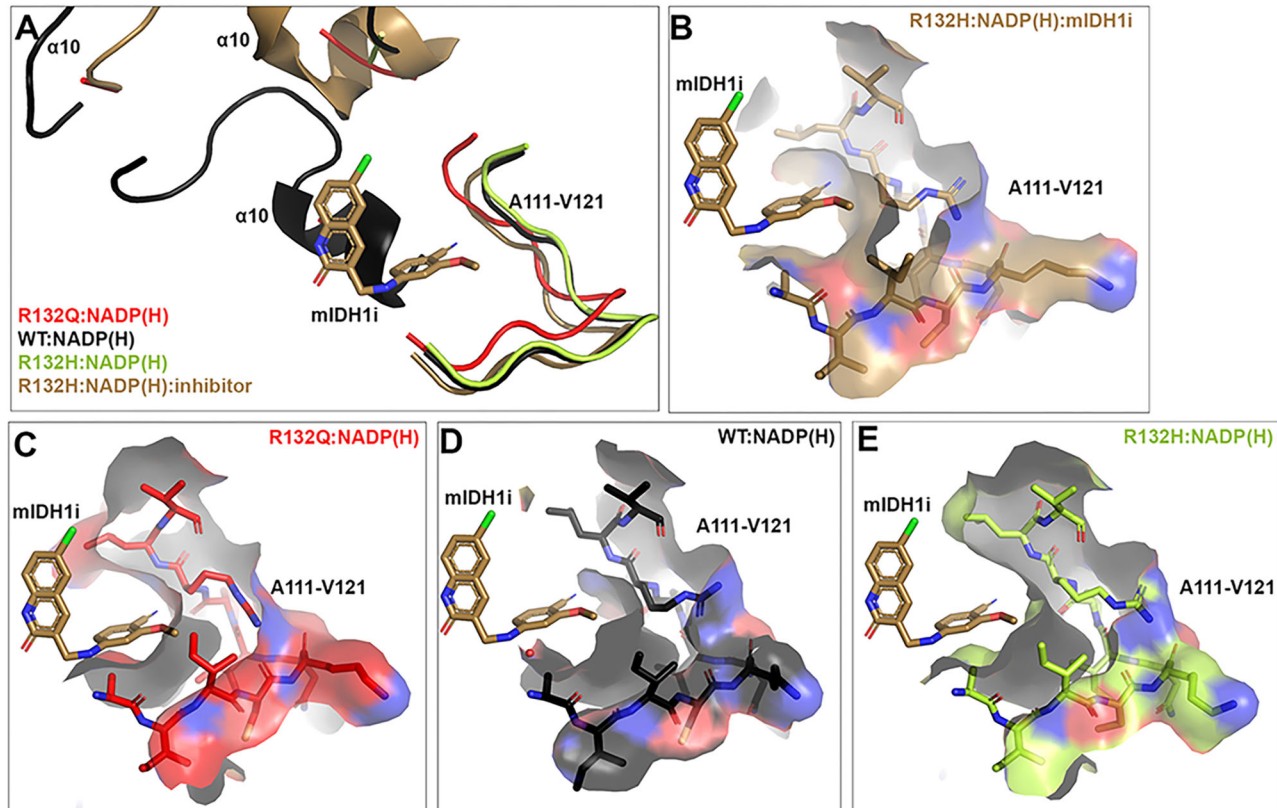

**Fig. 9 | Possible mechanisms of IDH1 R132Q selective mutant IDH1 inhibitor resistance.** We have reported previously that IDH1 R132Q binds selective mutant IDH1 inhibitors poorly. **A** A previously solved structure of a selective IDH1 R132H inhibitor (6O2Y)[40] was aligned to WT[13], R132H[30], and R132Q binary complexes. In (**B**–**E**), residues A111-V121 are shown as a surface. **B** The structure of the inhibitor bound to a R132H:NADP(H) complex[40]. Residues A111-V121 in R132Q:NADP(H) (**C**) and in WT:NADP(H) (**D**) obstruct the inhibitor binding pocket. **E** The inhibitor could be accommodated in the structure of R132H:NADP(H)[30].

X-100, and a protease inhibitor tablet), and cells were lysed using sonication. Crude lysates were clarified via centrifugation (12,000 rpm, 1 h, 4 °C). Lysate was loaded on to a pre-equilibrated Ni-NTA column and washed with 150 mL of wash buffer (20 mM Tris pH 7.5 at 4 °C, 500 mM NaCl, 15 mM imidazole, 5 mM BME), and protein was eluted with elution buffer (50 mM Tris pH 7.5 at 4 °C, 500 mM NaCl, 500 mM imidazole, 5% glycerol, 10 mM BME). Protein was dialyzed overnight in 50 mM Tris pH 7.5 @ 4 °C, 100 mM NaCl, 20% glycerol, and 1 mM DTT, and >95% purity was ensured via SDS-PAGE analysis. Finally, IDH1 protein was flash-frozen using liquid nitrogen and stored at −80 °C. All kinetic analysis was performed <1 month from cell pelleting.

For pre-steady-state kinetics and HDX-MS experiments, protein was loaded onto a pre-equilibrated (50 mM Tris-HCl 7.5 at 4 °C and 100 mM sodium chloride) Superdex 16/600 size exclusion column (GE Life Sciences, Chicago, IL) following Ni-NTA affinity chromatography to remove any protein aggregates. Protein was eluted with 50 mM Tris-HCl pH 7.5 at 4 °C, 100 mM NaCl, and 1 mM DTT. The fractions were pooled and concentrated for use in pre-steady-state experiments, or pooled and dialyzed in Tris-HCl pH 7.5 at 4 °C, 100 mM NaCl, 20% glycerol, and 1 mM DTT and used immediately for HDX-MS analysis[32]. For R132Q X-ray crystallography experiments, two 1 L cultures of terrific broth supplemented with 50 µg/ml of kanamycin were incubated at 37 °C and 180 rpm until an $A_{600}$ of 0.4 was reached. Cultures were removed and placed onto stir plates and allowed to cool to 25 °C. Expression was induced when cultures reached an $A_{600}$ of 0.8–1.0 with 1 mM IPTG and incubated for an additional 16–18 h. Cell pellets were harvested and resuspended in lysis buffer (20 mM Tris pH 7.5 at 4 °C, 500 mM NaCl, 0.2% Triton X-100, 5 mM imidazole, 1 mM PMSF, and 5 mM BME). Following cell lysis via sonication, crude lysate was clarified via centrifugation at 14,000 × g for one hour. The lysate was

loaded on to a pre-equilibrated Ni-NTA column. The column was washed with 100 mL of wash buffer (20 mM Tris pH 7.5 at 4 °C, 500 mM NaCl, 15 mM imidazole, 5 mM BME). Protein was eluted using elution buffer (50 mM Tris pH 7.5 at 4 °C, 500 mM NaCl, 500 mM imidazole, 5% glycerol, 10 mM BME). For the NADP(H)-stripped experiments, all required substrates for catalysis for the reverse (WT required MgCl₂, αKG, and bicarbonate) and neomorphic (R132H and R132Q required MgCl₂ and αKG) reactions were added to the Ni-NTA affinity column-bound IDH1 to convert any tightly binding NADPH to NADP⁺, followed by extensive column washing to remove the more weakly bound NADP⁺ as described in previous work[14]. In all cases, eluted protein was loaded onto a HiPrep 26/10 desalting column (GE Healthcare) containing 25 mM Tris pH 7.5 at 20 °C, 500 mM NaCl, 5 mM EDTA, 2 mM DTT, and placed on ice overnight to remove any remaining metals from purification. Fractions containing IDH1 were concentrated (MilliPore Amicon Ultra 15 30 kDa NMWL concentrator) and loaded onto a Superdex 26/600 (GE Healthcare) pre-equilibrated with 20 mM Tris pH 7.5 at 20 °C, 200 mM NaCl, and 2 mM DTT. Fractions containing pure IDH1 were pooled and concentrated to a final concentration of 14–20 mg/mL, flash frozen using liquid nitrogen, and stored at −80 °C. In all cases, the purity of the protein (>95%) was confirmed using SDS-PAGE analysis.

### Molecular graphics images
Structure figures were prepared using PyMOL v. 2.5.5[41].

### Kinetics assays
To measure steady-state activity of homodimer WT, R132H, and R132Q, only minor modifications were made from previous studies[21,22]. For the conventional reaction (ICT to αKG), IDH1 buffer (50 mM Tris HCl pH

7.5 at 37 °C, 150 mM NaCl, 10 mM MgCl₂, 1 mM DTT) and homodimer IDH1 (100 nM IDH1 WT, or 200 nM IDH1 R132H and R132Q), as well as various concentrations of ICT and 200 µM NADP⁺ were preincubated separately for 3 min at 37 °C. Following addition of substrates at 37 °C, the increase of absorbance at 340 nm due to production of NADPH was monitored using an Agilent Cary UV/Vis 3500 spectrophotometer (Santa Clara, CA). For the neomorphic reaction (αKG to D2HG), IDH1 buffer and homodimer mutant IDH1 (200 nM) as well as various concentrations of αKG at pH 7.5 and 200 µM NADPH were separately preincubated for 3 min at 37 °C. Following addition of substrates at 37 °C, the decrease of absorbance at 340 nm due to consumption of NADPH was monitored. As described previously[21,22], the kinetic parameters, which were obtained using two or three individual protein preparations (biological replicates), were determined by plotting the slope of the linear range of the change in absorbance over time. The change in absorbance was converted to nanomolar NADPH using the molar extinction coefficient for NADPH of 6.22 cm⁻¹ mM⁻¹ to determine $k_{obs}$ (i.e. nM NADPH/nM enzyme s⁻¹) at each substrate concentration. Each $k_{obs}$ was fit to the Michaelis-Menten equation using Graphpad Prism v.10 to calculate $k_{cat}$ and $K_m$, and technical replicate points are indicated.

For the reducing agent inhibition steady-state studies, the conventional reaction conditions described above were repeated except one of three reducing agents (DTT, TCEP, or BME) were added at varying concentrations during the pre-incubation step with the enzyme before substrates were added. Here, three protein preparations were used (biological replicates), with each point representing a single technical replicate. Upon obtaining Michaelis-Menten plots at various reducing agent concentrations, the inverse of both $k_{obs}$ and substrate concentration were plotted in Lineweaver-Burk analysis.

Single-turnover, pre-steady-state kinetic assays were performed for the neomorphic reaction at 37 °C using an RSM stopped-flow spectrophotometer (OLIS, Atlanta, GA). For the neomorphic reaction, hydride transfer (NADPH to NADP⁺ conversion) was monitored as a change in fluorescence as a function of time via measuring the depletion of NADPH signal by exciting the sample at 340 nm and scanning the emission spectrum from 410 to 460 nm. Final concentrations after mixing were as follows: 40 µM IDH1 R132Q or R132H, 10 µM NADPH, 10 mM αKG (IDH1 R132H) or 0.5 mM αKG (IDH1 R132Q), 50 mM Tris-HCl (pH 7.5 at 37 °C), 150 mM NaCl, 0.1 mM DTT, and 10 mM MgCl₂. The change in fluorescence as a function of time was fit to a single exponential equation ($Y = A_0 e^{-kt}$) using Graphpad Prism to obtain $k_{obs}$. For IDH1 R132H, a higher concentration of αKG (20 mM) was used since 1 mM αKG showed an initial lag.

Single turnover pre-steady-state kinetics were also performed for the conventional reaction at 37 °C to obtain rate constants associated with steps after NADP⁺ binding through hydride transfer using an RSM stopped-flow spectrophotometer. NADPH formation as a function of time was similarly monitored by exciting at 340 nm and scanning the emission spectrum from 410 to 460 nm. Final concentrations after mixing were as follows: 30 µM IDH1 WT or R132Q, 10 µM NADP⁺, 0.5 mM ICT (IDH1 WT) or 1 mM ICT (IDH1 R132Q), 50 mM Tris-HCl (pH 7.5 at 37 °C), 150 mM NaCl, 0.1 mM DTT, and 10 mM MgCl₂. The change in fluorescence as a function of time was fit to a single exponential equation ($Y = A_0 e^{-kt}$) using Graphpad Prism and $k_{obs}$ values were obtained.

Rates associated with NADPH binding corresponding to the first step of the catalytic cycle for the neomorphic reaction were performed using an RSM-stopped flow spectrophotometer (OLIS, Atlanta, Georgia). However, due to low sensitivity of our stopped-flow spectrophotometer, the concentrations of NADPH and IDH1 were increased, which in the case of IDH1 WT led to rates too fast to be detected by our instrument (≤100 s⁻¹). Therefore, glycerol (40%) and temperature (10 °C) were used to slow NADPH binding rates to IDH1. NADPH binding as a function of time was monitored by exciting at 340 nm and scanning the emission spectrum from 410 to 460 nm. Final concentrations after mixing were as follows: 4 µM IDH1, varying concentration of µM NADP⁺, 100 mM Tris-HCl pH 7.5, 150 mM NaCl, 0.1 mM DTT, 10 mM MgCl₂, and 40% glycerol. The change in fluorescence as a function of time was fit to a single exponential equation ($Y = A_0 e^{-kt}$) using Graphpad Prism, and $k_{obs}$ values were obtained and plotted as a function of NADPH concentration using the equation $k_{obs} = k_1[\text{NADPH}] + k_{-1}$. This yielded a linear graph indicating one-step binding, with the slope equal to $k_1$ and the Y-intercept equal to $k_{-1}$, though the Y-intercept slope was too high to do so reliably. For all pre-steady-state kinetics experiments except NADPH binding measurements, a single protein preparation was used with each trace representing an average of 4 technical replicates. For NADPH binding, 10 technical replicates were averaged. This work described here based on previously described experiments[14].

Isothermal titration calorimetry (ITC) experiments were conducted at the Sanford Burnham Prebys Protein Production and Analysis Facility using a Low Volume Affinity ITC calorimeter (TA Instruments). For NADPH titrations, experiments were performed at 25 °C in 20 mM Tris pH 7.5, 100 mM NaCl, 10 mM MgCl₂, and 2 mM BME, injecting 0.25 mM NADPH into the cell containing 0.025 mM for IDH1 WT, 0.025 mM or 0.04 mM IDH1 R132H; and injecting 0.15 mM NADPH into the cell containing 0.034 mM or 0.026 mM IDH1 R132Q. For ICT titrations, experiments were performed at 25 °C in 20 mM Tris pH 7.5, 100 mM NaCl, 10 mM CaCl₂, and 2 mM BME, injecting 0.6 mM ICT into the cell containing 0.12 mM IDH1 R132Q or 0.13 mM IDH1 R132H. Baseline control experiments were performed by injecting the ligand into a cell with buffer only. In all cases, ITC data were analyzed using the Nanoanalyze software package by TA Instruments.

## HDX-MS data collection and analysis

HDX-MS data collection and analysis was performed at the Biomolecular and Proteomics Mass Spectrometry Facility (BPMSF) of the University California San Diego using a Waters HDX-1 system which consists of a Leap PAL HTX-xt dual pipette autosampler controlled by HDxDirector software (v1.0.4.0) (Leap Technologies Inc, Carrboro, NC), which manages sample preparation for injection into a Waters UPLC HDX Manager temperature-controlled (0.1 °C) LC box (v1.50.1314), with solvent management by the combination of a Waters nanoAcquity UPLC Auxillary Solvent Manager (v1.50.2601) and a Waters nanoAcquity UPLC Binary Solvent Manager (v1.50.1327) delivering the sample into the Lockspray ESI source of a Waters Synapt G2-Si (UEA) quadrupole time-of-flight mass spectrometer, all Waters instruments being controlled by MassLynx 4.1 SCN917 (Waters Corporation, Milford, MA). Experiments were performed as previously described, using a sample of IDH1 WT without substrates to be analyzed alongside every experiment to allow direct experiment to experiment comparisons[32,42]. Deuterium exchange reactions were conducted using a Leap HDX PAL autosampler (Leap Technologies, Carrboro, NC). The D₂O buffer was prepared by lyophilizing sample buffer (50 mM Tris buffer at pH 7.5 at 4 °C, 100 mM NaCl, and 1 mM DTT) either alone (IDH1:NADP(H) condition) or with the following ligands: for the conventional reaction experiments, IDH1 WT and R132Q were treated with 0.01 mM NADP⁺ and 10 mM ICT (ternary complexes), or with 0.1 mM NADP⁺, 10 mM ICT, and 10 mM CaCl₂. For the neomorphic reaction experiments, IDH1 R132Q and R132H were treated with 0.1 mM NADPH and 10 mM αKG (ternary complexes), or with 0.1 mM NADPH, 10 mM αKG, and 10 mM CaCl₂ was also included (quaternary complexes). The buffer was first prepared in ultrapure water, lyophilized, and then redissolved in an equivalent volume of 99.96% D₂O (Cambridge Isotope Laboratories, Inc., Andover, MA) just prior to use. Deuterium exchange measurements were performed in triplicate for every time point (in the order of 0 min, 0.5 min, 1 min, 2 min, 5 min); each run took ~30 min to complete, including a blank run to ensure no carryover from run to run. Samples were prepared

~30 min prior to experimental setup and stored at 1 °C until dispensing into reaction vials at the start of the reaction, resulting in samples that were exposed to their substrates for between 2 h (0.5 min timepoint) and 7.5 h (last replicate of the 5 min timepoint) at 1 °C. IDH1 proteins were diluted to 5 μM in MS vials, to which the appropriate concentration of substrate(s) was added as indicated above (final volute of 150 μM), and this sample was placed in the 0.1 °C tray of the Leap autosampler. Sample (4 μL) alone or with substrate(s) were then removed by the autosampler from the original vial and transferred to a 25 °C tube in the other block of the autosampler, where they were equilibrated for 5 min at the reaction temperature (25 °C) before mixing with either $H_2O$ (control) or $D_2O$ buffer (56 μL) for the indicated times. Fifty μL of the $H_2O$- or $D_2O$-incubated sample was then transferred to a tube at 1 °C into which 50 μL 3 M guanidine hydrochloride had been pre-aliquoted (final pH 2.66). The sample was incubated for 1 min at 1 °C to quench deuterium exchange and denature the protein prior to injection of 90 μL of the sample into a 100 μL sample loop for in-line digestion at 15 °C using an immobilized pepsin column (Immobilized Pepsin, Pierce). Peptides were then captured on a BEH C18 Vanguard precolumn at 200 μL/min and then separated by analytical chromatography (Acquity UPLC BEH C18, 1.7 μm 1.0 × 50 mm, Waters Corporation) at 40 μL/min over 7.5 min using a 7–85% acetonitrile gradient containing 0.1% formic acid. The resultant elution was injected by electrospray into the Waters Synapt G2Si quadrupole time-of-flight mass spectrometer. Data were collected in the Mobility, ESI+ mode using a mass acquisition range of 200–2000 m/z, a scan time of 0.4 s, and the following settings: detector 2950 V, source temperature 80 °C, desolvation temperature 175 °C, sample cone 30 V, sample cone gas 50 L/h, desolvation gas 600 L/h, nebulizer gas 6.0 L/h. An infusion of leu-enkephalin (m/z = 556.277) every 30 s was used for continuous lock mass correction (mass accuracy of 1 ppm for calibration standard).

To identify peptides, data was collected on the mass spectrometer in mobility-enhanced data-independent acquisition (MS[E]), mobility ESI + mode. Peptide masses were determined from triplicate analyses, and resulting data were analyzed using the ProteinLynx global server (PLGS) version 3.0 (Waters Corporation). We identified peptide masses using a minimum number of 250 ion counts for low energy peptides and 50 ion counts for their fragment ions, with the requirement that peptides had to be larger than 1500 Da in all cases. Peptide sequence matches were filtered using the following cutoffs: minimum products per amino acid of 0.2, minimum score of 7, maximum MH+ error of 5 ppm, and a retention time RSD of less than 5%. To ensure high quality, we required that all peptides were present in two of the three experiments. After identifying peptides in PLGS, we then used DynamX 3.0.0 data analysis software (Waters Corporation) for peptide analysis. Here, relative deuterium uptake for every peptide was calculated via comparison of the centroids of the mass envelopes of the deuterated samples with non-deuterated controls per previously reported methods[43], and used to obtain data for coverage maps. Data are represented as mean values +/− SD of the three technical replicates due to processing software limitations, but we note that the LEAP autosampler robot provides highly reproducible data for biological replicates. Back-exchange was corrected for in the deuterium uptake values using a global back exchange correction factor (typically ~25%) determined from the average percent exchange measured in disordered termini of varied proteins[44] and validated through examination of highly disordered IDH1 peptides, with adjustment for slight differences in each experiment via comparison of the IDH1 WT without substrate control included in each experimental set. Given the very short run time (7 min total, 4 min window of peptide elutions), we have previously determined that a global correction based on fully exposed peptides suffices for back exchange correction[45]. Significance among differences in HDX data points was assessed using ANOVA analyses and t tests ($p$ value cutoff of 0.05) within DECA (v 116)[45] to determine a minimum significant difference of 0.25 Da for all peptides, as reported in the compliance table (Supplementary Table 5, Supplementary Fig. 19). Individual peptides of interest were compared via t-test of bound versus apo IDH1 using DECA[45] to validate significance. We generated deuterium uptake plots in DECA [github.com/komiveslab/DECA][45], with data plotted as deuterium uptake (back exchange-corrected) versus time. Deuterium uptake plots show the maximum possible deuterium uptake on the y axis, which is preferable to plotting the percent uptake or percent difference because it accounts for the size of the peptide. An HDX-MS data summary table is shown in Supplementary Table 5, and additional data are provided in Supplementary Data 1, 2, and 3.

## Crystallization

For the NADP(H)-only bound IDH1 R132Q crystals (PDB 8VHC, PDB 8VH9), enzyme (14–20 mg/mL) was incubated on ice with 10 mM NADPH. Crystals of R132Q:NADP(H) were grown via hanging drop vapor diffusion at 4 °C. 2 μL of IDH1 were mixed with 2 μL of well solution containing either 220 mM ammonium sulfate, 100 mM bis-tris pH 6.5, and 20% (w/v) PEG 3350 (PDB 8VHC), or well solution containing 200 mM ammonium citrate tribasic pH 7.0 and 26% (w/v) PEG 3350 (PDB 8VH9). Though both forms aligned very well and appeared otherwise identical, we feared the citrate buffer could nonetheless promote more substrate-bound-like features due to its structural similarity to isocitrate. Thus, the binary structure crystallized in sulfate was used for all further comparisons and alignments.

IDH1 R132Q crystals containing ICT (PDB 8VHD) were grown by first incubating the enzyme at 20 mg/mL with 10 mM NADP+, 10 mM $CaCl_2$, and 200 mM DL-isocitric acid at 20 °C for 1 h. Then, 2 μL of IDH1 were mixed with 2 μL of well solution containing 100 mM bis-tris propane pH 6.5, 200 mM NaI, and 24% (w/v) PEG 3350 and stored at 4 °C. Crystals were harvested using a nylon-loop and cryo-protected using a solution of 100 mM bis-tris propane pH 6.5, 200 mM NaI, 26%(w/v) PEG 3350, and 20%(v/v) glycerol. Crystals were flash-frozen in liquid nitrogen and stored until data collection.

IDH1 R132Q crystals containing αKG and/or αKG-adducts were generated by incubating enzyme (14–20 mg/mL) on ice with 10 mM NADPH, 20 mM $CaCl_2$, 75 mM αKG Fisher Scientific (Hampton, NH) for 1 h. For the PDB 8VHB structure, crystals were grown at 4 °C via hanging drop vapor diffusion, where 2 μL of IDH1 were mixed with 2 μL of the well solution containing 200 mM NaSCN and 21%(w/v) PEG 3350. Crystals were cryo-protected using a solution of 20% (v/v) glycerol, 25% (w/v) PEG 3350 and 200 mM NaSCN, and flash-frozen in liquid nitrogen and stored until data collection. For the PDB 8VHA structure, IDH1 R132Q was incubated at 20 °C with 10 mM NADPH, 10 mM $CaCl_2$, 10 mM αKG, and then crystals were grown at 4 °C by mixing 2 μL of IDH1 R132Q with 2 μL of well solution containing 160 mM $NaNO_3$ and 20% (w/v) PEG 3350. Crystals were harvested using a nylon-loop and cryo-protected in a solution containing 22% (v/v) glycerol and 26% (w/v) PEG 3350.

For IDH1 R132Q crystals containing the NADP-TCEP adduct (PDB 8VHE), enzyme (14–20 mg/mL) was incubated on ice with 10 mM NADP+, 20 mM $CaCl_2$, and 75 mM DL-isocitric acid for 1 h. Crystals were grown at 4 °C via hanging drop vapor diffusion, with 1.5 μL of IDH1 mixed with 1.5 μL of well solution containing 200 mM KSCN, 24% (w/v) PEG 6000, and 5 mM TCEP pH 7.4.

## Data collection, processing, and refinement

Data were collected at 100 K using synchrotron radiation at the Advanced Photon Source, beamline 24-ID-E or at the Stanford Synchrotron Radiation Lightsource, beamline BL12-2. All datasets were processed with XDS v6/30/23[46]. Structure solutions were obtained by molecular replacement using PHASER-MR in Phenix 1.2[47,48]. For αKG and/or αKG-adducts (PDB 8VHB, PDB 8VHA), isocitrate (PDB 8VHD), and NADP-TCEP (PDB 8VHE) co-crystals, PDB ensembles of PDB 1T0L[13],

PDB 4KZO[14], and PDB 6PAY[26] were used for molecular replacement by generating ensembles using Phenix Ensembler[47,48]. For IDH1 R132Q apo structures, PDB 1T09[13] and PDB 4UMX[49] were used as search models. The models were optimized via iterative rounds of refinement in Phenix Refine and manual rebuilding in Coot 1.1[50,51]. Ligand restraints were generated in Phenix eLBOW[47,48]. Data collection and refinement statistics are summarized in Supplementary Table 6, a stereo-image of the electron density maps for each structure reported here are shown in Supplementary Fig. 17, and mFo-DFc omit maps contoured at 3 sigma for all ligands are shown in Supplementary Fig. 18.

### Calculations

Density functional theory (DFT) calculations[52] were carried out to model the NADP-TCEP binding energetics and geometry using the Gaussian 16 vC.01 suite of programs[53]. The NADP$^+$ was modeled as the nicotinamide ring plus a pendant dihydroxy furan to represent the sugar. The model NADP$^+$ and NADP-TCEP adduct were each given a +1 charge. To better model the effects of the solvent, three explicit water molecules were included in calculations on the adducts, distributed at the likeliest sites for hydrogen bonding. The B3LYP[54], ωB97XD[55], and M06[56] hybrid functionals were used with the cc-pVDZ[57,58] and pc-$n$[59,60] basis sets, with the latter obtained from the online Basis Set Exchange[61]. In all of these calculations, implicit solvation was applied using the COSMO model with water as the solvent[62,63] and empirical dispersion was added using the D3 version of Grimme's dispersion along with Becke-Johnson damping[64,65]. This treatment of solvation effectively models the species as though they were in solution rather than crystalline form. Harmonic frequency analysis was carried out to obtain the vibrational corrections needed to calculate the free energies. Finally, because basis set superposition error can be substantial relative to intermolecular bond energies, the counterpoise correction was applied to our final energies of reaction[66,67]. The transition state (TS) for the TCEP + NADP$^+$ binding was identified and confirmed by analysis of the single imaginary vibrational frequency. The DFT calculations for the model NADP-TCEP adduct predicted values of 25° for $\Delta\theta_C$ and −11° for $\Delta\theta_N$, where the experimental values in the X-ray structure were $\Delta\theta_C = 29.2°$ and $\Delta\theta_N = −1.1°$ (Supplementary Table 2). For the NADP-αKG adduct, agreement was similar, with DFT predicting $\Delta\theta_C = 29°$ and $\Delta\theta_N = −14°$ as compared to $\Delta\theta_C = 25°$ and $\Delta\theta_N = −25°$ in the X-ray structure (Supplementary Table 2). The binding was energetically favored, and appeared to occur without barrier when vibrational effects were included, with a calculated binding energy of 9.4 kcal mol$^{-1}$ at 298 K. However, the calculated free energies indicated that in solution, the entropy decrease would preclude spontaneous binding. Quenching the translational entropy of the species in the crystal may be what allowed the process to occur. We noted that the counterpoise corrections to the transition state and adduct energies were essential, having magnitudes of 7–8 kcal mol$^{-1}$ and comparable to the uncorrected energy differences.

For the dihedral angles, the deviation from planarity $\Delta\theta$ of the NADP pyridine ring in the adduct was reported using the average of two dihedral angles. Numbering the carbon atoms in the ring by convention as shown in Supplementary Fig. 15, the C-P bond in NADP-TCEP formed at atom 4. The positions of the N atom 1 and the opposite C atom 4 are referenced to the plane defined by the roughly coplanar atoms 2, 3, 5, and 6. The average of the dihedral angles 2-3-5-4 and 6-3-5-4 (Supplementary Fig. 15) was subtracted from 180° to yield $\Delta\theta_C$ as a metric for the deviation from planarity of C4, while the average of 3-2-6-1 and 5-2-6-1 subtracted from 180° is used to calculate $\Delta\theta_N$ for N1. A sign convention was applied such that if $\Delta\theta_C$ and $\Delta\theta_N$ had the same sign, the two corners of the ring bend away each other in chair fashion, whereas opposite signs indicate a boat-like conformation. Comparison of the results from the different functionals and basis sets showed very little difference in the geometry. Optimized geometries obtained with

the pc-2 basis set on a smaller geometry (omitting sugar and explicit waters) were not significantly different from those obtained with pc-1, so we chose to report the B3LYP/pc-1 results here, with the sugar and explicit waters included (Supplementary Table 2). An additional geometry optimization was run on the NADP-αKG adduct with two explicit waters and a -2 charge, employing the aug-pc-1 basis set[59,68] to obtain the diffuse functions necessary to adequately model anions.

### Reporting summary

Further information on research design is available in the Nature Portfolio Reporting Summary linked to this article.

## Data availability

Crystallographic data and protein structure coordinates have been deposited with the Protein Data Bank (PDB) public repository: PDB 8VHC; PDB 8VH9; PDB 8VHD; PDB 8VHB; PDB 8VHA; and PDB 8VHE. Previously solved structures are also available: PDB 1T0L[13], PDB 4KZO[14], and PDB 6PAY[26]. Output files from the computational work are available at the ioChem-BD database [https://doi.org/10.19061/iochem-bd-6-320]. HDX-MS data can be found at the MassIVE FTP server [https://massive.ucsd.edu/ProteoSAFe/dataset.jsp?task=d24eb2fc5c0a4a2d9437dc1598212530]. Supplementary Information is included with Supplementary Figs. and Tables. Supplementary Data 1–3 with additional HDX-MS data are also included. Source Data is also included. Additional information and requests for resources and reagents should be directed for fulfillment by the corresponding author Christal D. Sohl (csohl@sdsu.edu). Source data are provided with this paper.

## Code availability

Deuterium uptake plots were generated using DECA, which can be accessed using the following link: github.com/komiveslab/DECA. For more details on the development of this code, please see the accompanying reference[45].

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

## Acknowledgements

IDH1 WT and R132H plasmids were obtained from Charles Rock (St. Jude's). This work was funded by a Research Scholar Grant, RSG-19-075-01-TBE, from the American Cancer Society (C.D.S.), National Institutes of Health R35 GM137773 (C.D.S.), MARC 1 T34 GM149430 (C.D.S.), MARC 5T34GM008303 (SDSU), and IMSD 5R25GM058906 (SDSU), as well as the California Metabolic Research Foundation (SDSU) and the Rees-Steely Research Foundation (E.A.). The HDX-MS core of the UCSD BPMSF is supported by NIH shared instrumentation grant S10 OD0016234. The Sanford Burnham Prebys Protein Production and Analysis Facility is supported by NCI Cancer Center Support Grant P30 CA030199. The Northeastern Collaborative Access Team beamlines are funded by NIH/NIGMS (P30GM124165) and the Eiger 16 M detector at the 24-ID-E beam line is funded by a NIH-ORIP HEI grant (S10OD021527). The Advanced Photon Source is a U.S. Department of Energy (DOE) Office of Science User Facility operated for the DOE Office of Science by Argonne National Laboratory under Contract No. DE-AC02-06CH11357. Use of the Stanford Synchrotron Radiation Lightsource, SLAC National Accelerator Laboratory, is supported by the U.S. DOE Office of Science, Office of Basic Energy Sciences under Contract No. DE-AC02-76SF00515. The SSRL Structural Molecular Biology Program is supported by the DOE Office of Biological and Environmental Research, and by NIH/NIGMS (P30GM133894). The content is solely the responsibility of the authors and does not necessarily represent the official views of the National Institutes of Health.

## Author contributions

M.M., N.A.S., D.A.M., E.A., B.M.C., A.A.B., A.L.C, and S.S. contributed to the methodology, data curation, data analysis, visualization, validation, and editing; R.K., T.M., N.J.C., K.A.S. contributed to the methodology, data curation, experimental analysis, and validation; E.A.K. contributed to funding acquisition, data analysis, and editing; J.M.S. contributed to the experimental analysis and editing; T.H. contributed to the conceptualization, data analysis, supervision, visualization, and editing; C.D.S. contributed to the conceptualization, data analysis, data curation, supervision, visualization, funding acquisition, writing, editing, and project administration.

## Competing interests

J.M.S. is an employee at Vividion Therapeutics and owns stock in Schrödinger. The remaining authors declare no competing interests.
