## [Peer Review File · Nature Communications]

Active site remodeling in tumor-relevant IDH1 mutants drives distinct kinetic features and potential resistance mechanismsREVIEWER COMMENTS

Reviewer #1 (Remarks to the Author):

In this work a biochemical, crystallography, mass spectrometry and computational study is presented on isocitrate dehydrogenase. The authors create several active site mutants where an Arg residue is replaced by another amino acid. This generally effects the reaction rates as shown from the experiments. They also provide crystal structures that give further insight into substrate binding. Overall the work is interesting and the topic fits the remit of Nature Communications.

1. I did not read the author instructions for this journal, but to me the paper looks very long for a communication. I suggest to shorten the paper. There are particularly large figures with structure drawings that I do not find very illustrative for the storyline (Figs 4, 5 and 6), which the authors may want to move to the SI.
2. Did the authors measure a solvent isotope effect?
3. Essentially, the authors mutate an amino acid involved with binding and positioning the substrate (an Arg residue) and replace it with a neutral amino acid that may not do that as well. It should not come as a surprise that the binding rates change. Some discussion is needed here.
4. I miss some general discussion on the results. What did we learn and how can this be applied to alternative systems?

Reviewer #2 (Remarks to the Author):

SUMMARY

In this work, the authors describe the study of human isocitrate dehydrogenase 1 (IDH1) WT, R132H and R132Q by using static and dynamic structural methods. The mechanism of IDH1 mutants is not well understood yet, and the authors use several techniques to give an insight into the mechanism of this protein. Overall, the manuscript details an interesting study based on a significant amount of work. However, some aspects of the data analysis and representation are not up to expected standards.

DETAILED COMMENTS

For most of the HDX analysis, the authors rely on absolute peptide deuterium uptake to support their conclusions. HDX is inherently variable, hence the authors should include the standard deviation of the measurements in all uptake plots (methods section states data have +/- SD but none of the plots seems to have the error bars).

Even though comparing total deuterium uptake of peptides in different protein states can give some information, is recommended to do differential experiments and a full statistical analysis of the data to properly identify if the changes between two states are significant (methods state they used ANOVA and t-tests to analyze the data, but nothing is found in the manuscript). As such, more information is needed to support the differences)

Line 144: Authors claim that peptides 210-216, 240-253 and 257-267 in IDH1 R132Q reached equilibrium faster than R132H, and that is supportive of R132Q being in a more closed conformation compared to R132H. Indeed, those peptides reach equilibrium faster, however if that pattern is common to all the peptides in the protein it wouldn't support that hypothesis. More clarification and potentially extra analysis would be required here.

Authors use HDX to analyze the several protein complexes. Binary, ternary and quaternary complexes of IDH1:NADP +/-ICT/aKG +/- Ca²⁺. Authors agree that IDH1 and its mutants have different affinity towards ICT/aKG, but they do not change the ligand concentration during HDX experiments to ensure maximum (or equally) complex formation. Authors should optimize ligand concentration.

HDX community recommendations: <https://www.nature.com/articles/s41592-019-0459-y>

In general, crystal structures are hard to “read”. Over imposing so many structures at the same time makes it difficult to identify differences between structures. We recommend the authors to try to represent the data in a simpler way.

MINOR ISSUES

Check description of ligands in legends (Figures 4 and 5).

P4, line 114, HDX experiments do not probe solvent accessibility.

Reviewer #3 (Remarks to the Author):

Comments:

This research article investigated the kinetic and structural characteristics of IDH1 R132Q mutant. Among the cancer-driven mutants of IDH1, the R132Q mutant is of special interests because it preserves the conventional IDH1 activity while acquiring neomorphic activity of D2HG production, and exhibits resistance to selective IDH1 mutant inhibitors. The authors conducted kinetic and HDX-MS experiments which suggest that compared to the R132H mutant, the R132Q mutant has a lower barrier to adopting an active conformation driven by substrate and metal binding. The authors also reported six IDH1 R132Q structures in different ligand-bound forms which support the kinetic data. A seatbelt model is proposed for ICT-bound, closed and catalytically relevant structures.

Significance: Understanding the molecular mechanism of the IDH1-R132Q mutant is of significance to development of more selective and effective inhibitors, and could also contribute to broader field of enzyme design.

Methodology: Most of the methodology is sound and the authors have provided enough details.

However, more replicates should be performed in the enzymatic assays in order to obtain reliable results.

Critique:

1. The cancer-driven mutations in IDH1 are typically heterozygous, resulting in the formation of the heterodimer of WT and mutant IDH1 (WT/mutant). In the case of R132H, the WT/R132H heterodimer shows different biochemical and structural properties compared to the R132H/R132H homodimer. This difference is an important consideration in inhibitor design. Is this also the case for the R132Q mutation? The authors should include some kinetic data for the WT/R132Q heterodimer and ideally the structure of the WT/R132Q heterodimer and compare them with the R132Q/R132Q homodimer.

2. Extended Data Fig. 1: “At least two protein preparations, indicated via different symbols, were used...Each point in the curve represents a single replicate.” – This is very unnormal and unacceptable. You cannot pick up different data points from two experiments and fit them into one saturation curve. To obtain reliable results, at least 2-3 independent experiments using different protein preparations should be performed, with each data point representing the average of the independent experiments. And the error bar for each data point should be shown.

3. Line 74: “IDH1 R132Q uniquely maintains modest catalytic efficiency for the conventional reaction” – the catalytic efficiency of the R132Q mutant is 180-fold lower than IDH1 WT, and only 5.5-fold higher than the R132H mutant. It would be more precise to describe the R132Q mutant as having weak catalytic efficiency for the conventional reaction.

4. Lines 104-109: What is the binding affinity of the R132Q or R132H mutant for α KG? Since α KG is the major substrate for these mutants, it would be valuable to determine the K_d for α KG and see if the results align with the kinetic and HDX-MS experiments.

5. Some of the structural figures are very crowded, especially so for Figures 4C, 4D, 5D, 5E and 5G. It is difficult to extract any useful information when four or more overall structures are superposed together. I would suggest to separate the structures into more panels. Also, presenting only the important structural elements instead of the overall structures would improve clarity.

6. Line 216: “300 fold” should be “31 fold”.

7. Lines 280-281: α KG can be both the substrate and product of IDH1 R132Q. Is it possible that α KG binds to the active site in different manners as a substrate or as a product? Could this be the reason why the α KG-bound R132Q structure does not appear in a catalytically relevant form?

8. The binding of NADP- α KG adduct in IDH1 R132Q is interesting. Is this NADP- α KG adduct formed in the enzyme active site as an intermediate during the catalytic reaction, or is it formed in the crystallization reagents first and then captured by the enzyme?

9. Lines 601-602: According to Supplementary Fig. 2, NADP- α KG adduct is formed from NADP⁺ and α KG.

However, in the crystallization experiments, NADP- α KG was observed when NADPH was used. This raises the question of how such an observation is possible. Can NADPH facilitate this reaction?

10. Lines 101-102: The presence of NADP(H) bound to the enzyme can significantly impact the results of NADPH binding and ITC experiments. Therefore, the author should assess the NADP(H) bound to the enzyme to ensure that the enzyme is free of cofactor. Simply stating that the enzyme was stripped of cofactor is insufficient.

11. Supplementary Table 6: In the data collection section, please provide the Rmerge values and the statistics for the highest resolution shell. In the refinement section, please provide the percentage of disallowed points in the Ramachandran plot. These parameters would help to evaluate the quality of the crystal structures.

Besides, the “unique reflections” parameters should be listed under data collection section, not the refinement section.

Also, please keep the effective digits consistent.

12. This article focuses on the conformational changes of the active site, while most IDH1 R132H inhibitors bind to the dimer interface. How does the His-to-Gln substitution affect the dimer interface and contribute to the R132Q mutant’s resistance against R132H inhibitors?

In the discussion, the author suggests that residues 111-121 (loop between the β 4 and β 5 strands) may contribute to the resistance. To clarify this point, it would be helpful to include a figure showing the conformational difference of this loop between R132Q and R132H structures.

Re: Response to reviewers regarding NCOMMS-24-08382A.

We are pleased that overall, reviewers found our work interesting and suitable for Nature Communications. We have addressed each point individually below in italics. We note that we have also reformatted the document to have the extended data in the supplementary information (the extended data was from a previous Nature journal submission), as well as general edits for clarity and conciseness. Reformatting related to comments below are highlighted in the manuscript in yellow.

Reviewer #1 (Remarks to the Author):

In this work a biochemical, crystallography, mass spectrometry and computational study is presented on isocitrate dehydrogenase. The authors create several active site mutants where an Arg residue is replaced by another amino acid. This generally effects the reaction rates as shown from the experiments. They also provide crystal structures that give further insight into substrate binding. Overall the work is interesting and the topic fits the remit of Nature Communications.

Response: We thank the reviewer for the support for our work.

1. I did not read the author instructions for this journal, but to me the paper looks very long for a communication. I suggest to shorten the paper. There are particularly large figures with structure drawings that I do not find very illustrative for the storyline (Figs 4, 5 and 6), which the authors may want to move to the SI.

Response: We have ensured that the manuscript is within required word limits. We agree that the structure figures (Figs. 4, 5) are challenging to read and can be improved. We have drastically simplified both figures and also generated two new Supplementary Figs. 9 and 10 that show less complex overlays. In the spirit of this comment, we also improved Supplementary Fig. 12. We have opted to keep Fig. 6 in the main text, as we feel this displays important information on the a10 helix from

both static and dynamic experiments. We appreciate the opportunity for improved clarity.

2. Did the authors measure a solvent isotope effect?

Response: Here we did not measure a solvent isotope effect. Hydride transfer has been identified as the rate-limiting step of typical IDH enzymes, and in our reported single turnover pre-steady-state experiments monitoring NADPH formation and consumption coupled to our steady-state experiments, we have shown that this step indeed likely remains the overall rate-limiting step.

3. Essentially, the authors mutate an amino acid involved with binding and positioning the substrate (an Arg residue) and replace it with a neutral amino acid that may not do that as well. It should not come as a surprise that the binding rates change. Some discussion is needed here.

Response: We thank the reviewer for the opportunity to clarify this point. We want to note that our selection of R132Q evolved from its tumor relevance as it has been noted as a very rare tumor-driving IDH1 mutation (eg Hirata, et al. PNAS (2015) 112, 2829-2834); and for the quite unique catalytic properties it has compared to all other tumor-driving IDH1 mutants (we reported on this in 2017 and 2018 as highlighted in the introduction). We added brief comments on this topic in the discussion section.

4. I miss some general discussion on the results. What did we learn and how can this be applied to alternative systems?

Response: We highlighted in the discussion how we pose a new selectivity handle for resistance mutations for mutant IDH1 inhibitor design. We added additional short discussion about using TCEP more broadly with dehydrogenases, and about predictions on differences between glutamine and histidine mutations based on their comparison to the structure of arginine as suggested in comment #3. We thank the reviewer for these opportunities to strengthen the manuscript.

Reviewer #2 (Remarks to the Author):

SUMMARY

In this work, the authors describe the study of human isocitrate dehydrogenase 1 (IDH1) WT, R132H and R132Q by using static and dynamic structural methods. The mechanism of IDH1 mutants is not well understood yet, and the authors use several techniques to give an insight into the mechanism of this protein. Overall, the manuscript details an interesting study based on a significant amount of work. However, some aspects of the data analysis and representation are not up to

expected standards.

Response: We thank the reviewer for the support for our work.

DETAILED COMMENTS

For most of the HDX analysis, the authors rely on absolute peptide deuterium uptake to support their conclusions. HDX is inherently variable, hence the authors should include the standard deviation of the measurements in all uptake plots (methods section states data have +/- SD but none of the plots seems to have the error bars). Even though comparing total deuterium uptake of peptides in different protein states can give some information, is recommended to do differential experiments and a full statistical analysis of the data to properly identify if the changes between two states are significant (methods state they used ANOVA and t-tests to analyze the data, but nothing is found in the manuscript). As such, more information is needed to support the differences)

Response: The reviewer raises an important point. We have included the +/- SD in the uptake plots, but the error is so small, it wasn't visible. To rectify this, we've decreased the symbol size in all the plots significantly so at least any larger error can be visible (close inspection will allow you to see some error bars now), though in general the error bars are often still too small to be visualized. The addition of +/- SD has also been added to the figure legends, and these numbers for all peptides have been added to the source data.

ANOVA and t-tests were performed in DECA to determine a minimum significant difference of 0.25 Da for all peptides, as reported in the compliance table (Supplementary Table 5).

Line 144: Authors claim that peptides 210-216, 240-253 and 257-267 in IDH1 R132Q reached equilibrium faster than R132H, and that is supportive of R132Q being in a more closed conformation compared to R132H. Indeed, those peptides reach equilibrium faster, however if that pattern is common to all the peptides in the protein it wouldn't support that hypothesis. More clarification and potentially extra analysis would be required here.

Response: We appreciate the opportunity to clarify this point. Uptake plots represent combined exchange rates for all amides in a particular protein. Thus, if a peptide has 13 amino acids, it is then a combination of 12 rates of exchange. However, exponential fitting can only ever capture 2-3 "rates" occurring at different orders of magnitude. For clarity, we have edited the text to reflect that we are having fewer amides exchanging at slower/intermediate/faster exchange rates vs "reaching equilibrium." (see our work in Mandell, et al, JMB (2000)).

Authors use HDX to analyze the several protein complexes. Binary, ternary and quaternary complexes of IDH1:NADP +/-ICT/aKG +/- Ca²⁺. Authors agree that IDH1 and its mutants have different affinity towards ICT/aKG, but they do not change the ligand concentration during HDX experiments to ensure maximum (or equally) complex formation. Authors should optimize ligand concentration.

Response: We note that the sample is prepared with a 20-fold excess of NADP⁺; however, it is then diluted 15-fold into deuteration buffer containing the same concentration of NADP⁺ (0.1 mM) for a final ratio of 300:1. As a result, the off-rate in the deuteration buffer should be negligible as demonstrated by the percent bound calculations now presented in the Methods. For the case of non-saturable binding the binding site is exposed instead of obscured, concomitantly what would be observed is deuteration that would otherwise be suppressed by the binding partner. Observing reduced deuteration in ligand-bound samples and, importantly, not observing distinct bimodals demonstrative of a partially bound state, is clear evidence that binding is saturated.

HDX community recommendations: <https://www.nature.com/articles/s41592-019-0459-y>

In general, crystal structures are hard to “read”. Over imposing so many structures at the same time makes it difficult to identify differences between structures. We recommend the authors to try to represent the data in a simpler way.

Response: We agree that the structure figures (Figs. 4, 5) are challenging to read and can be improved. We have drastically simplified both figures and also generated two new Supplementary Figs. 9 and 10 that show less complex overlays. In the spirit of this comment, we also improved Supplementary Fig. 12. We appreciate the opportunity for improved clarity.

MINOR ISSUES

Check description of ligands in legends (Figures 4 and 5).

Response: We have checked all figure legends carefully, and all is correct. We included aKG-bound structures in Fig. 4 and ICT-bound structures in Fig. 5 overlays to show differences among the R132Q forms. These figures have now been moved to the Supplementary information (see above).

P4, line 114, HDX experiments do not probe solvent accessibility.

Response: HDX-MS methods have been shown to be an effective method to probe solvent accessibility by measuring deuterium uptake. We have published extensively in this area in the past, including, most relevant to this comment, the following: Accurate Prediction of Amide Exchange in the Fast Limit Reveals Thrombin Allostery. Markwick PRL, Peacock RB, Komives EA. Biophys J. 2019 Jan 8;116(1):49-56. doi: 10.1016/j.bpj.2018.11.023.

Reviewer #3 (Remarks to the Author):

Comments:

This research article investigated the kinetic and structural characteristics of IDH1 R132Q mutant. Among the cancer-driven mutants of IDH1, the R132Q mutant is of special interests because it preserves the conventional IDH1 activity while acquiring neomorphic activity of D2HG production, and exhibits resistance to selective IDH1 mutant inhibitors. The authors conducted kinetic and HDX-MS experiments which suggest that compared to the R132H mutant, the R132Q mutant has a lower barrier to adopting an active conformation driven by substrate and metal binding. The authors also reported six IDH1 R132Q structures in different ligand-bound forms which support the kinetic data. A seatbelt model is proposed for ICT-bound, closed and catalytically relevant structures.

Significance: Understanding the molecular mechanism of the IDH1-R132Q mutant is of significance to development of more selective and effective inhibitors, and could also contribute to broader field of enzyme design.

Methodology: Most of the methodology is sound and the authors have provided enough details. However, more replicates should be performed in the enzymatic assays in order to obtain reliable results.

Response: We thank the reviewer for the support for our work.

Critique:

1. The cancer-driven mutations in IDH1 are typically heterozygous, resulting in the formation of the heterodimer of WT and mutant IDH1 (WT/mutant). In the case of R132H, the WT/R132H heterodimer shows different biochemical and structural properties compared to the R132H/R132H homodimer. This difference is an important consideration in inhibitor design. Is this also the case for the R132Q mutation? The authors should include some kinetic data for the WT/R132Q heterodimer and ideally the structure of the WT/R132Q heterodimer and compare them with the R132Q/R132Q homodimer.

Response: We appreciate the opportunity for clarification. We have focused this manuscript solely to homodimer data (steady-state and pre-steady-state kinetics, HDX-MS, and crystallography all report on homodimer formation). To date, the crystallographic focus has been on IDH1 homodimers, with one non-inhibitor-bound heterodimer structure available (PDB 3MAS, bound to ICT and NADP(H)) that has poor density near the region of mutation and the regulatory domain. To allow comparison with WT and mutant structures, and to avoid the possibility of a complex equilibrium of WT:WT, Mutant:Mutant, and WT:Mutant dimer populations in our kinetic and HDX-MS work, we have focused all work here on WT:WT and Mutant:Mutant homodimers.

2. Extended Data Fig. 1: “At least two protein preparations, indicated via different symbols, were used...Each point in the curve represents a single replicate.” – This is very unnormal and unacceptable. You cannot pick up different data points from two experiments and fit them into one saturation curve. To obtain reliable results, at least 2-3 independent experiments using different protein preparations should be performed, with each data point representing the average of the independent experiments. And the error bar for each data point should be shown.

Response: we apologize for potentially misleading the reviewer – we did indeed design these experiments with high rigor, as each plot represented two biological replicates -- two independent experiments each taken from a unique protein preparation to ensure batch-to-batch consistency. We purposefully select unique concentrations from each prep to have high point density along the Michaelis-Menten curve, and plotted the results of these two biological replicate experiments on a single graph to generate the single Michaelis-Menten curve. As you note, this means we did not have replicates for individual points. One strategy with the data we have is to separate out the two plots and find independent kinetic parameters and then report the error in the difference of these calculated values. A second strategy is exactly what you describe here, which we have now included (we now show either three or four biological replicates for our steady-state work that include two technical replicates to make up the mean shown with SEM error bars). To do so, we generated one additional protein preparation for WT and R132H enzymes, and two additional protein preparations for R132Q enzyme and repeated previously used concentrations and displayed these with error bars as suggested (Supplementary Fig. 1). We appreciate the opportunity to make our work better.

3. Line 74: “IDH1 R132Q uniquely maintains modest catalytic efficiency for the conventional reaction” – the catalytic efficiency of the R132Q mutant is 180-fold lower than IDH1 WT, and only 5.5-fold higher than the R132H mutant. It would be more precise to describe the R132Q mutant as having weak catalytic efficiency for the conventional reaction.

Response: We agree that this is more precise phrasing and have altered the text accordingly.

4. Lines 104-109: What is the binding affinity of the R132Q or R132H mutant for α KG? Since α KG is the major substrate for these mutants, it would be valuable to determine the K_d for α KG and see if the results align with the kinetic and HDX-MS experiments.

Response: We had attempted to measure the K_d for α KG for both R132H and R132Q using the gold standard technique of isothermal titration calorimetry (ITC, Sanford Burnham Prebys core facility). We found that NADPH was necessary to facilitate measurements, but of course, this can lead to turnover, complicating ITC measurements. To prevent catalysis but allow α KG binding, we added Ca^{2+} , and measured K_d values of $8 \mu M$ for R132Q, and overall weak binding for R132H ($\sim 600 \mu M$). The stoichiometry was high for R132Q (3.9) and potentially for R132H, though the latter had weak enough binding where several stoichiometric fits were possible. In addition, the titration curves for R132Q had multiphasic shapes. Because of these issues and the potential for multiple interpretations, we did not include these data. We ultimately determined the stoichiometric issues were likely due to Ca^{2+} not quite fully being able to inhibit R132Q, leading to small but measurable turnover that produced enough heat to affect our measurements. To overcome this, we tested inhibition with very large metals in the same group number (for eg., strontium inhibited R132Q well), but ultimately concluded this would also have notable structural and ligand binding consequences making any K_d measurements less meaningful. We also monitored intrinsic protein fluorescence in pre-steady-state kinetics experiments to see if there was any signal associated with α KG in an effort to measure k_{on} and k_{off} rates to determine K_d (as described for NADPH in the manuscript), but we this step appeared spectroscopically silent.

5. Some of the structural figures are very crowded, especially so for Figures 4C, 4D, 5D, 5E and 5G. It is difficult to extract any useful information when four or more overall structures are superposed together. I would suggest to separate the structures into more panels. Also, presenting only the important structural elements instead of the overall structures would improve clarity.

Response: We agree that the structure figures (Figs. 4, 5) are challenging to read and can be improved. We have drastically simplified both figures and also generated two new Supplementary Figs. 9 and 10 that show less complex overlays. In the spirit of this comment, we also improved Supplementary Fig. 12. We appreciate the opportunity for improved clarity.

6. Line 216: “300 fold” should be “31 fold”.

Response: We appreciate your spotting this typo. We've made the correction in the text.

7. Lines 280-281: α KG can be both the substrate and product of IDH1 R132Q. Is it possible that α KG binds to the active site in different manners as a substrate or as a product? Could this be the reason why the α KG-bound R132Q structure does not appear in a catalytically relevant form?

Response: We appreciate this clever insight; this is an interesting possibility. We have adjusted the text to include this idea.

8. The binding of NADP- α KG adduct in IDH1 R132Q is interesting. Is this NADP- α KG adduct formed in the enzyme active site as an intermediate during the catalytic reaction, or is it formed in the crystallization reagents first and then captured by the enzyme?

Response: We were very curious about this, too, and had considered using our WT TCEP and WT/mutant DTT data to extrapolate a hypothesis in the discussion. We've expanded this examination in the discussion.

9. Lines 601-602: According to Supplementary Fig. 2, NADP- α KG adduct is formed from NADP⁺ and α KG. However, in the crystallization experiments, NADP- α KG was observed when NADPH was used. This raises the question of how such an observation is possible. Can NADPH facilitate this reaction?

Response: The reviewer brings up a good point. NADPH is likely not facilitating this adduct formation. Thiocyanate, which we used as an additive in our α KG-containing crystal conditions, has been shown to be capable of oxidizing NAD(P)H (Hogg and Jago (1970) 117(4), 791-7, and thus we suspect that NADP⁺ resulted from oxidation of NADPH by thiocyanate, which then reacted with α KG to form the adduct. Others have also reported adduct formation with NADP⁺ upon incubation of crystals with NADPH, similarly citing auto-oxidation (Paidimuddala, et al. FEBS (2018) 285, 4445-64. We've added this information to the figure legend to prevent similar concerns by readers.

10. Lines 101-102: The presence of NADP(H) bound to the enzyme can significantly impact the results of NADPH binding and ITC experiments. Therefore, the author should assess the NADP(H) bound to the enzyme to ensure that the enzyme is free of cofactor. Simply stating that the enzyme was stripped of cofactor is insufficient.

Response: we agree this is critical. The standard protocol for achieving stripped IDH1 (first described by Rendina, et al Biochemistry 2013, 52, 4263-4577) is to bind the

protein to the nickel column, add all required substrates for catalysis for the reverse (WT; MgCl₂, αKG, and bicarbonate) and neomorphic (R132H, R132Q; MgCl₂ and αKG,) reactions to convert any tightly binding NADPH to NADP⁺, and then wash the column extensively (100+ column volumes) to ensure full removal of any NADP⁺, which binds with much lower affinity. We then ensure no activity by adding ICT or αKG and metal but no cofactor. This information can be found in the cited paper, but we also expanded the information in the methods.

11. Supplementary Table 6: In the data collection section, please provide the Rmerge values and the statistics for the highest resolution shell. In the refinement section, please provide the percentage of disallowed points in the Ramachandran plot. These parameters would help to evaluate the quality of the crystal structures. Besides, the “unique reflections” parameters should be listed under data collection section, not the refinement section. Also, please keep the effective digits consistent.

Response: We thank the reviewer for the opportunity to improve the manuscript. We have updated this Supplementary Table as suggested here, including consistent significant digits.

12. This article focuses on the conformational changes of the active site, while most IDH1 R132H inhibitors bind to the dimer interface. How does the His-to-Gln substitution affect the dimer interface and contribute to the R132Q mutant’s resistance against R132H inhibitors? In the discussion, the author suggests that residues 111-121 (loop between the β4 and β5 strands) may contribute to the resistance. To clarify this point, it would be helpful to include a figure showing the conformational difference of this loop between R132Q and R132H structures.

Response: We appreciate the opportunity to improve this figure. We have shown in previous work that purified R132Q protein and cells expressing R132Q are resistant to mutant IDH1 inhibitors (similar to WT, only a pan-IDH inhibitor was effective, reported in Avellaneda Matteo, et al Biochem J (2018) 474, 3221-3238). We had attempted to highlight this region in the first version, but we realize we fell short of this goal when we failed label this region in the figure. We’ve improved this figure to make it clearer the differences between the two mutants as suggested by the reviewer using a different orientation and labeling. We also brought it forward to the main document.

REVIEWERS' COMMENTS

Reviewer #2 (Remarks to the Author):

The authors have made a good effort revising the manuscript. However a few points still remain:

a) Regarding the point that largely absolute peptide deuterium uptakes were used in the manuscript to support their conclusions, the authors may want to justify why they did not carry out differential experiments throughout the manuscript?

b) The authors should provide details for all HDX-MS experiments in the form of tables as recommended by the HDX community paper: <https://www.nature.com/articles/s41592-019-0459-y>

Reviewer #3 (Remarks to the Author):

The authors have largely addressed most of my comments in the revision. Nevertheless, they failed to address the issue about the heterodimer of WT and mutant IDH1 which is the cancer-driven heterozygous IDH1 found in patients (comment #1). Considering the technical difficulty in preparing the WT:mutant heterodimer, I would suggest the authors to discuss this issue in Discussion, pointing out that the mutant:mutant homodimer is an artificial model and the obtained results from the mutant:mutant homodimer may not fully represent the WT:mutant heterodimer.

Reviewer #2: a) Regarding the point that largely absolute peptide deuterium uptakes were used in the manuscript to support their conclusions, the authors may want to justify why they did not carry out differential experiments throughout the manuscript? b) The authors should provide details for all HDXMS experiments in the form of tables as recommended by the HDX community paper:
<https://www.nature.com/articles/s41592-019-0459-y>

Reviewer #2: a) We have included a description and justification of generation of uptake plots in the methods. b) HDX summary table recommended in the Masson et al paper is found in our Supplementary Table 5. All details (and additional ones) are included. This was included in the original and revision versions.

Reviewer #3 (Remarks to the Author): The authors have largely addressed most of my comments in the revision. Nevertheless, they failed to address the issue about the heterodimer of WT and mutant IDH1 which is the cancerdriven heterozygous IDH1 found in patients (comment #1). Considering the technical difficulty in preparing the WT:mutant heterodimer, I would suggest the authors to discuss this issue in Discussion, pointing out that the mutant:mutant homodimer is an artificial model and the obtained results from the mutant:mutant homodimer may not fully represent the WT:mutant heterodimer

Reviewer #3: The following has been added to the manuscript results: We note that all of these mutant structures (both R132H and R132Q) describe mutant:mutant homodimers; with these mutations found heterozygously in patients, a possibility exists for WT:mutant heterodimers as well, which could result in still different structural features and conformations.”